# Budgeting Counterfactual for Offline RL

**Yao Liu**[1], **Pratik Chaudhari**[1,2], **Rasool Fakoor**[1]
[1]Amazon Web Services, [2]University of Pennsylvania
{yaoliuai,prtic,fakoor}@amazon.com

## Abstract

The main challenge of offline reinforcement learning, where data is limited, arises from a sequence of counterfactual reasoning dilemmas within the realm of potential actions: What if we were to choose a different course of action? These circumstances frequently give rise to extrapolation errors, which tend to accumulate exponentially with the problem horizon. Hence, it becomes crucial to acknowledge that not all decision steps are equally important to the final outcome, and to budget the number of counterfactual decisions a policy make in order to control the extrapolation. Contrary to existing approaches that use regularization on either the policy or value function, we propose an approach to explicitly bound the amount of out-of-distribution actions during training. Specifically, our method utilizes dynamic programming to decide where to extrapolate and where not to, with an upper bound on the decisions different from behavior policy. It balances between the potential for improvement from taking out-of-distribution actions and the risk of making errors due to extrapolation. Theoretically, we justify our method by the constrained optimality of the fixed point solution to our $Q$ updating rules. Empirically, we show that the overall performance of our method is better than the state-of-the-art offline RL methods on tasks in the widely-used D4RL benchmarks.

## 1 Introduction

One of the primary hurdles in reinforcement learning (RL), or online RL, is its reliance on interacting with an environment in order to learn [47]. This can be a significant barrier to applying RL to real-world problems, where it may not be feasible or safe to interact with the environment directly [48]. In contrast, batch or offline RL [6, 27, 30] provides a more suitable framework for effectively learning policies by leveraging previously collected data to learn a policy. Notably, offline RL relies on a fixed but limited dataset comprising previously collected data from an unknown policy(s) and lacks the ability to continue interacting with the environment to gather additional samples.

These limitations in offline RL give rise to various challenges, with one notable issue being the occurrence of extrapolation problems resulting from the scarcity of training data [8, 12]. In order to overcome this challenge, previous approaches to offline RL primarily focus on constraining the gap between the behavioral policy and the learned policy [4, 11, 12, 25, 40, 51], or limiting the disparity between the state-action values of logged actions in the data and extrapolated actions [8, 22, 26, 50, 53]. Alternatively, other approaches utilize a learned model that avoids making predictions outside the distribution of the dataset [21, 35, 55]. All these offline RL methods involve a sequence of counterfactual reasoning problems within the realm of potential actions, explicitly or implicitly. The question of "what if" we were to choose a different course of action than the behavior policy often arises, leading to extrapolation errors. Hence the difficulty of the problem increases as we plan for a longer horizon and involves more counterfactual decisions. For offline RL algorithms, it is difficult to find a balance between the potential for improvement from taking out-of-distribution actions and the risk of making errors due to extrapolation.

37th Conference on Neural Information Processing Systems (NeurIPS 2023).

A critical aspect of offline RL which is relatively unexplored pertains to the varying importance of different actions at each step in determining the final outcome. Not all actions are created equal, and some have more influence on the outcome than others. Thus, it is not always essential to consider alternative actions or counterfactuals at every step than the one dictated by the behavior policy. Rather, the emphasis should be placed on the actions that wield the most impact, warranting careful decision-making at those steps. This limits the number of counterfactual decisions we make and assigns them to the most needed states/steps, to increase the return on the "extrapolation investment".

Unlike existing approaches that employ explicit or implicit regularization on either the policy, value functions, or both, we introduce a novel and effective algorithm, called *Budgeting Counterfactual for Offline RL* (BCOL), that explicitly constrains the level of extrapolation during training. Specifically, our approach leverages dynamic programming to determine when and where extrapolation should occur, with a budget on the deviation from the behavior policy. This enables us to strike a balance between the potential for improvement achieved by exploring out-of-distribution actions and the inherent risks associated with extrapolation errors. Conceptually, such a method is different from regularized offline RL by allowing non-uniform constraints while keeping strict bounds on counterfactual decisions.

**Our contribution.** We propose a novel algorithm, BCOL, making only a few but important counterfactual decisions in offline RL. This idea extends the space of current offline RL methods that extensively focus on regularized methods. We demonstrate that the fixed point resulting from our $Q$-value updating rule corresponds to the optimal $Q$-value function, under the constraints that the policy deviates from the behavior policy within a budget. We conduct thorough evaluations on a diverse array of tasks from the widely-utilized D4RL benchmarks [9]. In terms of overall performance, our approach exhibited a favorable comparison to existing state-of-the-art methods.

## 2 Problem Settings

We study the RL problem in the context of the infinite horizon, discounted Markov Decision Process (MDP) [41]. An MDP is defined as a tuple $\mathcal{M} = <\mathcal{S}, \mathcal{A}, r, P_0, P, \gamma>$ where $\mathcal{S}$ is a state space, $\mathcal{A}$ is an action space, and both of these spaces can be infinite or continuous in nature. $r : \mathcal{S} \times \mathcal{A} \to \mathbb{R}$ is the reward function. $P_0$ is a distribution over $\mathcal{S}$ that the initial states are drawn from. $P : \mathcal{S} \times \mathcal{A} \to \Delta(\mathcal{S})$ maps a state-action pair to a distribution over state space. $\gamma$ is the discount factor over future reward. The goal in MDP is to maximize the expectation of discounted future reward $v^\pi = \mathbb{E}\left[\sum_{t=0}^{\infty} \gamma^t r_t \mid a_t \sim \pi\right]$. An important function in MDP is the state-action value function, known as the $Q$ function, which represents the expected future reward, given an initial state or state-action pair. $Q^\pi(s, a) := \mathbb{E}\left[\sum_{t=0}^{\infty} \gamma^t r_t \mid s_0 = s, a_0 = a\right]$.

The primary emphasis of this paper is on model-free learning algorithms within the context of offline RL. The goal is to learn a target policy $\pi$ that maximizes the expected future reward using a fixed dataset $\mathcal{D}$. The dataset $\mathcal{D}$ consists of transitions $(s, a, r, s', a')$ where $s$ can be drawn from any fixed distribution, $a \sim \mu(\cdot|s)$, $r = r(s, a)$, $s' \sim P(\cdot|s, a)$, $a' \sim \mu(\cdot|s')$. Here $\mu$ denotes an unknown behavior policy that was utilized in the past to collect dataset $\mathcal{D}$.

In offline RL, the task of learning an optimal policy poses a greater challenge compared to online RL. This is primarily due to the fact that the target policy can be different from the behavior policy, leading to the introduction of out-of-distribution actions. This challenge is further amplified by the use of function approximation and the problem horizon as the extrapolation errors from fitted $Q$ function tend to accumulate over time-steps [8, 12]. In particular, it will results in the $Q(s, a)$ values increasing dramatically for out-of-distribution state and action pairs. Consequently, this results in the learning of a policy that is likely to perform poorly and risky at the deployment time. Many previous works on offline RL aim to address these issues by some form of regularization, which might hurt the flexibility of making certain key decisions at each step (which we will return to later). However, we take a different approach in this paper where we emphasize on cautiousness of making counterfactual decisions and, hence propose a method that only deviates from behavior policy for a few steps.

## 3 Method

As mentioned earlier, prior studies in offline RL mainly address the issue of extrapolation through two types of regularization terms: policy regularization or value regularization. Policy regularization

typically involves utilizing a divergence metric between current candidate policy and behavior policy while value regularization is commonly defined as the positive difference between $Q(s, \hat{a})$[1] and $Q(s, a)$, where $a \sim \mu(\cdot|s)$. In these works, the regularization terms were *uniformly* applied to all samples in the dataset, enforcing a flat penalization on the distance between $\pi(\cdot|s)$ and $\mu(\cdot|s)$, or flat penalization of overestimating $Q(s, \hat{a})$ for all $s$. The utilization of a uniform regularization approach can present a problem. When strong regularization is enforced, the divergence bound from behavior policy over all states are small, hence the policy may not be able to improve significantly. In such cases, the resulting policy can only be as good as the behavior policy itself. Conversely, due to the fact that trajectory distribution mismatch increase exponentially with respect to the problem horizon [8, 12, 34, 36], if the bound is not small enough, it causes significant issues such as large extrapolation errors that lead to a risky and poor-performing policy. Therefore, applying regularization uniformly without considering the specific characteristics of each state can pose challenges and limitations in offline RL.

This work emphasizes the importance of making *counterfactual decisions*, i.e., decisions that are different from the decisions that would have been made by the behavior policy. Building upon the notion that not all decision steps carry equal importance, we introduce a novel offline RL algorithm that incorporates only a limited number of counterfactual decisions. In other words, the algorithm follows the behavior policy $\mu$ most of the time, but it only makes counterfactual decisions (i.e. performing offline RL) in certain states. Thus intuitively it maximizes the potential improvement from counterfactual decisions while keeping the extrapolation error under control[2]. We refer to this number by the *budget* of counterfactual, denoted as $B$. The grid world MDP in Figure 1 shows a simple example of why using a budget of counterfactuals is an effective constraint.

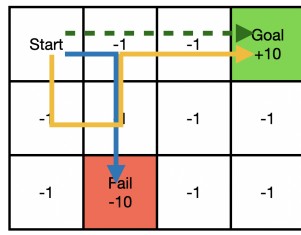

**Figure 1:** Grid world example

Here, the goal is to find the shortest path from the start to the goal state without passing through the fail state, given a dataset of previously recorded trajectories from different and unknown behavior policies. There are two types of trajectories in the dataset, marked as yellow and blue in Figure 1. While yellow trajectories are the most common, the shortest path requires stitching together these two types of trajectories and taking a counterfactual decision (shown by dashed green line). A budget of counterfactuals ensures that the agent only takes the most beneficial counterfactual decision and prevents it from taking unnecessary counterfactual actions.

In order to spend the budget of counterfactual strategically, a dynamic programming algorithm is utilized to plan and asses the potential policy improvement resulting from actions taken outside of the data distribution. Thus, the choice of policy on each step needs to balance between the immediate gain from taking greedy actions in the current step and the benefit from future counterfactual decisions. Naturally, the policy also depends on how many counterfactual decisions it can take before it exceeds the upper bound $B$. We define the budget of counterfactual decisions at a time step $t$ as follows:

$$b_{t+1} = b_t - \mathbb{1}\{\pi(\cdot|s_t, b_t) \neq \mu(\cdot|s_t)\}, \quad b_0 = B, \tag{1}$$

where $\mathbb{1}$ is an indicator function. The policies studied in this paper take the current budget $b_t$ as an input, which $b_t$ is the initial budget $B$ subtracted by the number of counterfactual steps taken before time step $t$ as shown in (1). Given that, the goal of our method is to solve the following constrained policy optimization:

$$\max_{\pi} \mathbb{E}\left[\sum_{t=0}^{\infty} \gamma^t r_t \mid s_0 \sim P_0, \pi\right] \; s.t. \, b_t \geq 0, \; \forall t \geq 0, \; \forall \{(s_t, b_t, a_t)\}_{t=0}^{\infty} \in \mathcal{E}_{\mathcal{M}}(\pi) \tag{2}$$

$\mathcal{E}_{\mathcal{M}}(\pi)$ is the support set of trajectory distribution introduced by the MDP and policy $\pi$. Without additional statements, later we only consider state-action-budget trajectories in $\mathcal{E}_{\mathcal{M}}(\pi)$ and drop this

---

[1]There are different approaches to obtaining $\hat{a}$ [5, 8]. Generally, it approximates $\text{argmax}_a Q(s, \cdot)$ or is sampled from the current policy.

[2]Technically, extrapolation refers to out-of-distribution actions. A counterfactual decision policy may still overlap with the behavior policy and thus not truly extrapolate. Whether or not actions from counterfactual policy lead to extrapolation is unknown when we only observe logged action rather than $\mu$. In this paper, we refer to this "possible extrapolation" as extrapolation as well, for the simplicity of discussion.

requirement for the ease of notation. Note that $b_t$ in (2) is a function of the initial budget $b_0 = B$, policies, and states before step $t$. To provide more clarity, we expand the constraint in (2) as follows:

$$\sum_{t=0}^{\infty} \mathbb{1}\{\pi(\cdot|s_t, b_t) \neq \mu(\cdot|s_t)\} \leq B \tag{3}$$

In order to develop our offline RL algorithm that maximizes (2), we introduce a new Bellman operator that plans on backup future values as well as the number of counterfactual decisions.

**Definition 1 (Counterfactual-Budgeting Bellman Operator).**

$$\mathcal{T}_{\text{CB}} Q(s, b, a) := r(s, a) + \gamma \, \mathbb{E}_{s'} \left[ V_Q(s', b) \right] \tag{4}$$

$$\text{where } V_Q(s', b) := \begin{cases} \max\{\max_{a'} Q(s', b - 1, a'), \, \mathbb{E}_{a' \sim \mu} Q(s', b, a')\} & b > 0 \\ \mathbb{E}_{a' \sim \mu} Q(s', b, a') & b = 0 \end{cases}$$

The Bellman operator $\mathcal{T}_{\text{CB}}$ updates the $Q$ values by taking the maximum value between two terms. The first term refers to the case where a counterfactual decision is made by maximizing $Q$ values over all possible actions in the next state. This leads to a decrease in the counterfactual budget. The second term involves following the behavior policy in the next state (similar to SARSA [43]) while keeping the counterfactual budget intact. By selecting the optimal backup from the two cases, the $Q$ value for a given budget $b$ strikes a balance between the benefits of counterfactual decisions (i.e. $\max_{a'} Q(s', b-1, a')$) in the next step and in further future. Since $b \leq B$, there are at most $B$ backup steps taking the maximum $\max_{a'} Q(s', b - 1, a')$ in the backup path. This intuitively upper bounds the amount of extrapolation that the $Q$ function can take. We can recover the standard Bellman operator on state-action Q functions by considering $b \equiv \infty$ in $Q(s, b, a)$ and the rule $\infty - 1 = \infty$. Since $b = b - 1 = \infty$, the first max operator in $V_Q$ always prefers the former term and gives us the standard Bellman operator.

It is straightforward to show the counterfactual-budgeting Bellman operator is a $\gamma$-contraction. Based on that, we prove the fixed point is the optimal $Q$ function constrained by a counterfactual budget.

**Theorem 2.** There exists a unique fixed point of $\mathcal{T}_{\text{CB}}$, and it is[1]

$$Q^{\star}(s, b, a) := \max_{\pi} \mathbb{E} \left[ \sum_{t=0}^{\infty} \gamma^t r_t \mid s_0 = s, a_0 = a, b_1 = b, \pi \right], \quad s.t. \ b_t \geq 0, \forall t \geq 1 \tag{5}$$

This theorem (proved in the appendix) indicates that the fixed point iteration of $\mathcal{T}_{\text{CB}}$ will lead to the optimal value function with the upper bound of counterfactual decisions. Thus it motivates us to minimize a temporal difference (TD) error of the counterfactual-budgeting Bellman operator, towards the goal of only improving behavior policy over a limited number of but important decision steps.

## 3.1 Algorithm

In this section, we derive a practical offline algorithm for our approach when the counterfactual-budgeting Bellman operator (4) is estimated via function approximation. First, we need to replace the expectations in (4) by one-sample estimation to evaluate the Bellman backup with the fixed dataset $\mathcal{D}$. Next, considering that the action space is continuous, we approximate the $\max_a$ operator in (4) by maximizing over actions sampled from a policy network that is trained to maximize the current $Q$ function. Note that these design choices are commonly employed in previous offline RL algorithms [8, 12, 25, 26]. As a result, we obtain a sampled-based counterfactual-budgeting Bellman operator denoted as $\widehat{\mathcal{T}}_{\text{CB}}$.

**Definition 3 (Approximate Counterfactual-Budgeting Bellman Operator).**

$$\forall(s, a), \, \widehat{\mathcal{T}}_{\text{CB}} Q_{\theta}(s, b, a) := r(s, a) + \gamma \begin{cases} \max\{\max_{\bar{a} \in \{a_k\}_{k=1}^m} Q_{\theta}(s', b - 1, \bar{a}), \, Q_{\theta}(s', b, a')\} & b > 0 \\ Q_{\theta}(s', b, a') & b = 0 \end{cases}$$

---

[1]Here $Q^{\star}(s, b, a)$ is defined as given $b_1 = b$ rather than $b_0$ because the action $a$ is already given and the future (optimal) value should be independent to which distribution $a$ is drawn from.

| **Algorithm 1** BCOL Training | **Algorithm 2** BCOL Inference |
|---|---|
| 1: **Input:** $\theta^0,\ \phi^0,\ T,\ B,\ \mathcal{D}$ | 1: **Input:** $\theta,\ \phi,\ B,\ \widehat{\mu},\mathcal{M}$ |
| 2: **for** $t = 0$ **to** $T-1$ **do** | 2: $b_0 \leftarrow B,\ s_0 \sim \mathcal{M}$ |
| 3: $\quad \theta^{t+1} \leftarrow \theta^t - \alpha_q \nabla_\theta \mathcal{L}_Q(\theta, \theta^t; \phi^t, B, \mathcal{D})$ | 3: **for** $t = 0$ **to** trajectory ends **do** |
| 4: $\quad \phi^{t+1} \leftarrow \phi^t - \alpha_p \nabla_\phi \mathcal{L}_\pi(\phi; \theta^t, B, \mathcal{D})$ | 4: $\quad \widehat{\pi}, b_{t+1} \leftarrow \texttt{Select}(\pi_\phi, \widehat{\mu}; s_t, b_t, Q_\theta)$ |
| 5: **end for** | 5: $\quad a_t \sim \widehat{\pi}(\cdot),\ r_t, s_{t+1} \sim \mathcal{M}$ |
| 6: **Return** $\theta^T, \phi^T$ | 6: **end for** |

where $(s', a')$ sampled from $\mathcal{D}$ and $\{a_k\}_{k=1}^m$ are sampled from $\pi$. Although this operator requires additional input $b$ to the $Q$ functions compared with the standard Bellman operator, it does not add any extra data requirement. We can augment the $(s, a, s', a')$ tuple from an offline dataset with all $0 \le b \le B$ where $B$ is the maximum number of counterfactual decisions that we consider.

To learn the $Q$ function, we minimize the least square TD error introduced by $\widehat{\mathcal{T}}_{\text{CB}}$, using dataset $\mathcal{D}$ consisting of $(s, a, s', a')$ sampled from behavior policy $\mu$. Subsequently, we update the policy by making it more likely to choose actions with higher $Q$-values. The objective functions for the $Q$ function and the policy $\pi$ are formally defined as follows, with $\theta$ representing the parameters of the $Q$ function and $\phi$ representing the parameters of the policy networks:

$$\mathcal{L}_Q(\theta, \bar{\theta}; \phi, B, \mathcal{D}) := \sum_{b=0}^{B} \mathbb{E}_{(s,a,s',a') \sim \mathcal{D}} \left[ \left( Q_\theta(s, b, a) - \widehat{\mathcal{T}}_{\text{CB}} Q_{\bar{\theta}}(s, b, a) \right)^2 \right] \tag{6}$$

$$\mathcal{L}_\pi(\phi; \theta, B, \mathcal{D}) := -\sum_{b=0}^{B} \mathbb{E}_{s \sim \mathcal{D}, a \sim \pi_\phi(\cdot|s,b)} Q_\theta(s, b, a) \tag{7}$$

where $\bar{\theta}$ denotes delayed target network. Algorithm 1 describes the training process of our method[1]. The counterfactual-budgeting Bellman operator has an interesting property that $Q$ is monotonically increased with $b$, for any $Q$ such that $Q = \widehat{\mathcal{T}}_{\text{CB}} Q$.

$$Q(s, b, a) \ge Q(s, b', a), \quad \forall s, a, b > b'. \tag{8}$$

The proof is straightforward since $Q$ with a larger budget maximizes the value over more action sequences. This property will be followed asymptotically. However, with a limited number of iterations, and function approximations, the gap is not always positive. Intuitively, enforcing the monotonic gap might reduce the search space and potentially accelerate the convergence. It is important to note that this modification does not alter the fixed point solution since $Q^\star$ always has a monotonic gap. Such a regularizer does not enforce any behavior/pessimistic constraint to the policy. It only constrains the Q-values for the same action with different budgets, not the Q-values for different actions. This regularizer will help the dynamics programming over counterfactual actions finding a self-consistent solution.

To implement it, we introduce the following penalty term and add it to the $\mathcal{L}_Q$. This results in the revised form of $\mathcal{L}_Q$ as follows:

$$\mathcal{L}_Q(\theta, \bar{\theta}; \phi, B, \mathcal{D}) + \omega \sum_{b=0}^{B-1} \mathbb{E}_{s \sim \mathcal{D}, a \sim \pi_\phi(\cdot|s,b)} \left[ (\max\{Q_\theta(s, b, a) - Q_\theta(s, b+1, a), 0\})^2 \right] \tag{9}$$

This penalty term minimizes the gap for the actions sampled from $\pi_\phi(\cdot|s, b)$. As $\pi_\phi(\cdot|s, b)$ maximizes $Q_\theta(s, b, \cdot)$, which is supposed to be smaller than $Q_\theta(s, b+1, \cdot)$, it can be viewed as a more efficient sampler of actions, compared to uniformly sampling, in order to ensure the constraints in Equation 8.

Putting these together, we have Algorithm 1: Budgeting Counterfactual for Offline reinforcement Learning (BCOL). It uses an offline actor-critic style algorithm to learn the $Q$ function and a policy maximizing the learned Q values. However, as the problem is to find the best policy under the constraints, greedy policy $\pi_\phi$ itself is not enough for inference. The action selection method during inference is implicitly included in the definition of $\mathcal{T}_{\text{CB}}$. At test time, the policy needs to look ahead based on the current budget $b_t$ and the $Q$ values of taking counterfactual actions v.s. taking the

---

[1]Note $\mathcal{L}_\pi$ is not exact but abstract in the sense that gradient to the $\phi$ as a distribution parameter cannot be calculated directly and need the so-called policy gradient estimator. We clarify our implementation in Section 3.3.

behavior policy. Based on that decision, we updated the new budget as well. We define this step as an operator below.

$$\texttt{Select}(\pi, \mu; s, b, Q) := \begin{cases} (\mu(\cdot|s), b) & \text{if } \mathbb{E}_{\bar{a}\sim\pi(s,b)} Q(s, b-1, \bar{a}) \leq \mathbb{E}_{\bar{a}\sim\mu(s)} Q(s, b, \bar{a}) \text{ or } b = 0 \\ (\pi(\cdot|s, b), b-1) & \text{o.w.} \end{cases}$$

The complete inference time procedure is described in Algorithm 2. It also takes the learned policy and $Q$ function parameters as well as an approximate behavior policy $\widehat{\mu}$. In case of unknown behavior policy, $\widehat{\mu}$ can be learned from behavior cloning or any other imitation learning algorithm. Algorithm 2 starts with an initial counterfactual budget $B$, takes action each time according to the condition in Select, and update the budget $b_t$ if the action is not drawn from $\widehat{\mu}$.

### 3.2 Comparison to regularized and one-step offline RL

One of the most used methods in offline RL methods is adding policy or value regularization and constraints terms on top of a vanilla off-policy RL algorithm [5, 8, 12, 22, 23, 25, 26, 50, 51], referred to as regularized methods in this paper. Our method can be viewed as an alternative to using regularized losses. Instead, we enforce a non-Markov budget constraint on the policy class, and we argue this provides several unique advantages. First, compared with the coefficients of regularization terms, the budget parameter has a more clear physical explanation in most applications. Thus it is easier to tune the hyper-parameters and explain the decisions from the policy in practice. Second, the constraints on budget will never be violated in the test, it provides an additional level of safety. While regularization terms penalize the divergence during training, they cannot provide a guarantee on the divergence or distance between test policy and behavior policy.

Another type of offline RL that is related to our core idea is one-step RL [3, 49]. We propose and leverage the concept of limiting the number of counterfactual/off-policy steps. However, the "step" here refers to the decision steps, and in one-step RL it refers to the (training) iteration step. Although one-step RL only applies only one training step, the resulting policy can still be far away from the behavior policy in many different states, and give different decisions during the test. From another perspective, our method, although with a limited number (or even one) of counterfactual decision steps, still applies dynamic programming to find the best allocation of counterfactual steps.

### 3.3 Important implementation details

Algorithm 1 provided a general actor-critic framework to optimize policy value constrained by counterfactual decisions. To implement a practical offline deep RL algorithm, the first design choice is how we model the policy and choose the policy gradient estimator to $\nabla_\phi \mathcal{L}_\pi$. Algorithm 1 is compatible with any off-policy policy gradient methods. We implement Algorithm 1 using SAC-style [17] policy gradient for stochastic policy and TD3-style [13] policy gradient for deterministic policy, to provide a generic algorithm for both stochastic and deterministic policy models. As TD3 uses deterministic policy, the value of $m$ is 1 in $\widehat{\mathcal{T}}_{\text{CB}}$ and the actor loss becomes $Q_\theta(s, b, \pi_\phi(s, b))$. In both cases, we implement target updates following SAC/TD3. We defer more details to the appendix.

Both SAC and TD3 are originally proposed for online, off-policy RL. To better fit into the offline setting, previous offline RL methods based on these algorithms equip them with several key adaptations or implementation tricks. To eliminate orthogonal factors and focus on the budgeting idea in algorithm comparison, we follow these adaptations and describe them below. For SAC, prior work use twin $Q$ functions [26], and a linear combination of the two $Q$ values [8]. Prior work [8, 24] also sample $m$ actions from the actor and take the maximum $Q$ values in backup, instead of a standard actor-critic backup, which is a trick firstly introduced in [12, 15]. The entropy term in SAC is dropped in [8] as this term is mostly for online exploration. Previous TD3-based work [11] normalizes the state features and $Q$ losses in TD3. All these adaptations are commonly used in state-of-the-art offline RL and are mostly considered as minor implementation details rather than major algorithmic designs. We refer to SAC and TD3 with all these adaptations as offline SAC and offline TD3 in this paper and refer to the two families of algorithms SAC-style and TD3-style.

One-step RL[3] and IQL [23] for offline RL is built on top of estimating the behavior value $Q^\mu$ by SARSA. Unlike SAC or TD3, it is not applicable to apply the counterfactual budgeting idea on top of SARSA, since it is a native on-policy method and does not consider counterfactual decisions. Thus we focus on the implementation of BCOL with SAC and TD3.

| Task Name | BC | 10%BC | DT | RAMBO | ARMOR | IQL | Onestep | TD3 +BC | BCOL (TD3) | CQL | CDC | BCOL (SAC) |
|---|---|---|---|---|---|---|---|---|---|---|---|---|
| halfcheetah-m | 42.6 | 42.5 | 42.6 | **77.6** | 54.2 | 47.4 | 55.6 | 48.4 | 45.0 | 46.1 | 62.5 | 50.1 |
| hopper-m | 52.9 | 56.9 | 67.6 | 92.8 | **101.4** | 66.3 | 83.3 | 59.4 | 85.8 | 64.6 | 84.9 | 83.2 |
| walker2d-m | 75.3 | 75.0 | 74.0 | 86.9 | **90.7** | 78.3 | 85.6 | 84.5 | 76.7 | 74.5 | 70.7 | 84.1 |
| halfcheetah-mr | 36.6 | 40.6 | 36.6 | **68.9** | 50.5 | 44.2 | 42.5 | 44.4 | 40.9 | 45.4 | 52.3 | 46.2 |
| hopper-mr | 18.1 | 75.9 | 82.7 | 96.6 | 97.1 | 94.7 | 71.0 | 50.1 | 83.4 | 92.3 | 87.4 | **99.8** |
| walker2d-mr | 26.0 | 62.5 | 66.6 | 85.0 | **85.6** | 73.9 | 71.6 | 80.2 | 49.7 | 83.7 | **87.8** | 86.0 |
| halfcheetah-me | 55.2 | 92.9 | 86.8 | **93.7** | 93.5 | 86.7 | 93.5 | 91.5 | 88.7 | 87.3 | 66.3 | 86.9 |
| hopper-me | 52.5 | 110.9 | 107.6 | 83.3 | **103.4** | 91.5 | 102.1 | 100.5 | 106.7 | 109.2 | 83.2 | 99.0 |
| walker2d-me | 107.5 | 109.0 | 108.1 | 68.3 | **112.2** | 109.6 | 110.9 | 110.1 | 108.5 | 109.9 | 103.9 | **110.9** |
| mujoco total | 466.7 | 666.2 | 672.6 | 753.1 | **788.6** | 692.4 | 716.0 | 669.2 | 685.6 | 713.0 | 699.0 | 746.0 |
| antmaze-u | 54.6 | 62.8 | 59.2 | 25.0 | - | 87.5 | 64.3 | **96.3** | 93.3 | 94.0 | 93.6 | 90.3 |
| antmaze-u-d | 45.6 | 50.2 | 53.0 | 16.4 | - | 62.2 | 60.7 | 71.7 | 68.0 | 47.3 | 57.3 | **90.0** |
| antmaze-m-p | 0.0 | 5.4 | 0.0 | 0.0 | - | **71.2** | 0.3 | 1.7 | 12.3 | 62.4 | 59.5 | 70.0 |
| antmaze-m-d | 0.0 | 9.8 | 0.0 | 0.0 | - | 70.0 | 0.0 | 0.3 | 14.0 | **74.3** | 64.6 | 72.3 |
| antmaze-l-p | 0.0 | 0.0 | 0.0 | 23.2 | - | **39.6** | 0.0 | 0.0 | 0.0 | 34.2 | 33.0 | 35.6 |
| antmaze-l-d | 0.0 | 6.0 | 0.0 | 2.4 | - | **47.5** | 0.0 | 0.3 | 0.0 | 40.7 | 25.3 | 37.6 |
| antmaze total | 100.2 | 134.2 | 112.2 | 67.0 | - | 378.0 | 125.3 | 171.3 | 187.7 | 352.9 | 333.5 | **396.0** |
| **Total** | 566.9 | 800.4 | 784.8 | 820.1 | - | 1070.4 | 841.3 | 840.2 | 873.3 | 1065.9 | 1032.5 | **1142.0** |

**Table 1:** Average normalized scores on D4RL tasks. Task names for MuJoCo: m=medium, mr=medium replay, me=medium expert. Task names for antmaze: u=umaze, m=medium, l=large, p=play, d=diverse. **Bold numbers** stand for the globally best and underlined numbers stand for the best with in the base method (SAC or TD3) group.

Looping over all budget values in the loss is not efficient in practice. We implement the $Q$ function and policy with $B$ output heads and tensorize the loop over $b$. One can further reduce the computation cost by sampling budget values instead of updating all $B$ heads over every sample.

# 4   Experiments

We evaluate our BCOL algorithm against prior offline RL methods on the OpenAI gym MuJoCo tasks and AntMaze tasks in the D4RL benchmark [9]. We compare the SAC-style and TD3-style implementation of BCOL with state-of-the-art offline RL algorithms, studying the effectiveness of budgeting counterfactual in offline RL. Our experiment results also reveal that behavior cloning with only one strategic counterfactual decision still work surprisingly well on MuJoCo tasks. Finally, we study the value of our dynamic programming methods on where to spend the counterfactual budget.

**Baselines.** We compared BCOL with regularized methods (policy and/or value regularization) based on SAC and TD3: CQL [26], CDC [8], TD3+BC [11]. Besides regularized methods, we also compare with one-step RL method [3] and IQL [23] as other state-of-the-art model-free offline RL methods, RAMBO [42] and ARMOR [52] as state-of-the-art in model-based offline RL, and behavior cloning, behavior cloning on the top $10\%$ data, and decision transformer as imitation learning baselines. This covers a set of strong offline RL baselines. We are interested in both studying the effectiveness of counterfactual budget ideas in contrast to the closest regularized methods and state-of-the-art offline RL methods. We present a set of representative and the most performant offline RL baselines in this section, and defers the comparison against more offline RL baselines results to the Appendix due to the limit of space.

**Benchmark.** We report the results of BCOL together with baselines on 9 OpenAI gym MuJoCo tasks and 6 AntMaze tasks in D4RL. The gym MuJoCo tasks consist of the v2 version of medium, medium-reply, and medium-expert datasets in halfcheetah, walker2d, and hopper. MuJoCo tasks generally prefer imitation-type algorithm, as it contains a large portion of near-optimal trajectories. The 6 AntMaze tasks are considered harder tasks for offline RL as they contain very few or no near-optimal trajectories, and many previous methods lack performance reports on these tasks. We exclude the random datasets in MuJoCo tasks as they are not less discriminating and none of the offline RL algorithms learns a meaningful policy there. We exclude expert datasets in MuJoCo tasks since they can be solved simply by behavior cloning and mismatches the consideration in most offline RL algorithm designs.

**How we report the results.** Due to the inconsistency in the version of D4RL in the literature, it is necessary to clarify the source of baseline results here. We retrieve the results of baseline methods from the original papers if applicable (IQL, CQL, one-step RL, DT, RAMBO, ARMOR). We report the Rev. KL Reg variant of one-step RL as it shows the best performance on MuJoCo tasks [3]. For AntMaze tasks we report one-step RL and decision transformer results from the IQL paper as they

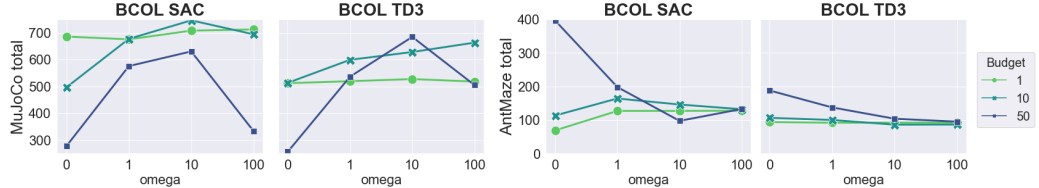

**Figure 2:** Total normalized score with different values of $B$ and $\omega$ in BCOL. The left two plots show MuJoCo average scores and the right two plots show AntMaze average scores.

are not reported in the original paper. We report the scores of behavior cloning and $10\%$ behavior cloning. We report the scores of behavior cloning from the IQL paper. We report the scores of CQL from an updated version of CQL paper[1] for D4RL v2 environments. We report the score of TD3+BC and CDC from our implementation since the original results are for D4RL v0 environments. The v2 results are generally better than the original v0 scores in the original papers and are consistent with TD3+BC's v2 scores from the IQL paper.

Table 1 shows the results of baseline algorithms as well as two variants of BCOL (TD3-style and SAC-style) on 15 D4RL tasks. For the scores from our experiment (BCOL , CDC, TD3+BC), we report the test episodic reward averaged over 300 test episodes as 10 test episodes per evaluation, the last 10 evaluations, and 3 runs with different random seeds. All algorithms run with 1M steps following the literature. We defer details like learning curves and standard deviations in the appendix.

**Main results.** As Table 1 shows, BCOL-SAC outperforms prior offline RL methods for both MuJoCo and AntMaze total scores. BCOL never falls behind the best methods by a large margin in any task. This indicates the effectiveness and robustness of the counterfactual budget in different scenarios. Especially for the total score of AntMaze, the harder tasks, BCOL-SAC outperforms most prior approaches by a large margin except IQL. We also find for both TD3-style and SAC-style algorithms, BCOL outperform the regularized method based on the same architecture, ablating orthogonal factors other than the key idea about the counterfactual budget.

**Hyper-parameters: $B$ and $\omega$.** Our algorithm only adds two hyper-parameters on top of SAC/TD3: budget $B$ and $\omega$. We searched the value of $B$ in $\{1, 10, 50\}$ and the value of $\omega$ in $\{0, 1, 10, 100\}$. We select one set of hyper-parameters for MuJoCo (SAC-style: $B = 10, \omega = 10$, TD3-style: $B = 50, \omega = 10$) and one set for AntMaze (SAC-style and TD3-style: $B = 50, \omega = 0$) based on the overall performance. This provides a fair comparison with baseline methods as all of them select their hyper-parameters either as this, or per task.[2] Figure 2 shows how the total scores changes with different values of $B$ and $\omega$. Full results with scores on each task are in the appendix.

Figure 2 also shows another surprising finding from our experiments on the MuJoCo tasks. With the budget being only one $B = 1$, BCOL (SAC) shows comparable performance to some recent offline RL work including CDC and IQL. With 10 steps out of 1000 steps, it is able to outperform most prior methods. This highlights how strategically adding a few important actions on top of behavior policy can provide strong performance in this benchmark. It also indicates that the MuJoCo tasks in D4RL, even the more mixed datasets, considerably prefer the imitation-type approach.

**Ablation on how we realize the counterfactual budgeting constraints.** The idea of budgeting counterfactual decisions naturally requires both the planning on budgeting in Algorithm 1 during the training and the planning on budgeting in Algorithm 2 during the testing. Ablation on the roles of budgeting itself and two planning parts is nonetheless helpful for us to understand how BCOL works. In Figure 3, we report the results of three ablation methods.

The first column shows the performance without budgeting constraints. This leads to offline TD3 or SAC algorithm with all implementation adaptations used by BCOL and others [11, 12, 26]. As we expected, vanilla offline RL without budgeting does not work well. The second column shows the performance with budgeting but without any planning on budgeting during either the training or the testing. This method realizes the budgeting by randomly assigning $B$ decisions to the policy

---

[1]https://sites.google.com/view/cql-offline-rl

[2]TD3+BC paper does not include results and hyperparameters in AntMaze. We do the hyper-parameter search of their $\alpha$, within the range provided in the paper, and report the highest total score with $\alpha = 3$.

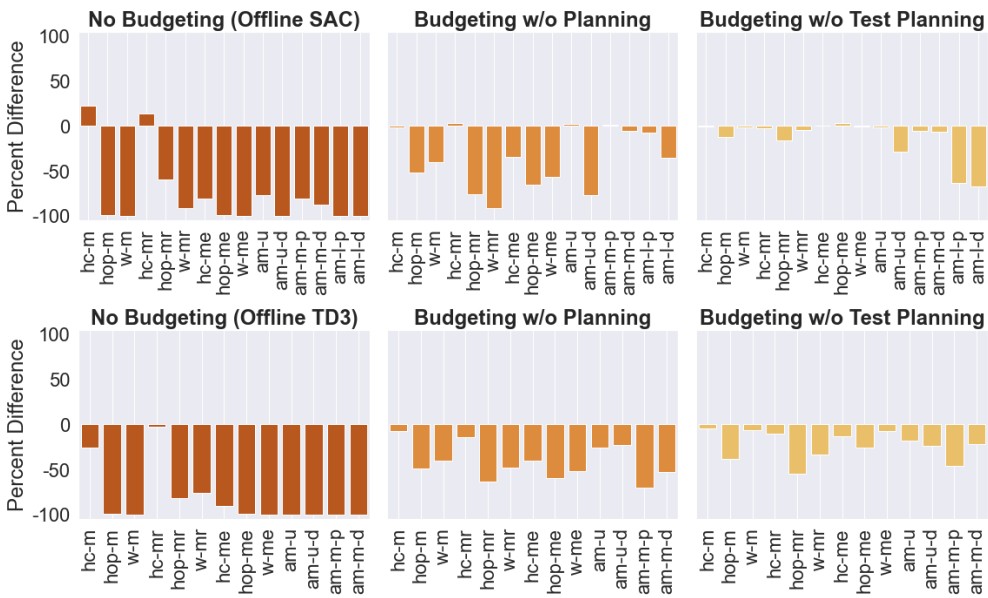

**Figure 3:** Percent difference of the performance on different budgeting methods compared with the full BCOL Algorithm (hc = HalfCheetah, hop = Hopper, w = Walker2d, am=AntMaze). The top row shows SAC-based experiments and the bottom row shows TD3-based experiments. TD3 plots do not include AntMaze-large tasks since the performances of BCOL are zero. No budgeting stands for offline SAC/TD3 without the budgeting constraints (equivalent to $B \to \infty$). Budgeting without planning stands for randomly selecting $B$ steps to follow from $\pi$ and the rest from $\hat{\mu}$ during the test, where $\pi$ is learned by offline SAC/TD3. Budgeting without test-time planning stands for randomly selecting $B$ steps (uniformly within the max horizon) to follow from $\pi$ and the rest from $\hat{\mu}$ during the test, where $\pi$ is learned by Algorithm 1. In all settings, $B$ is the same value as selected by BCOL .

learned by offline SAC/TD3, and the rest to the estimated behavior policy. Randomly stitching the counterfactual policy and behavior policy fails to fully realize the benefit of budgeting, since both training and inference are not budget-aware. The experiment in the third column studies if it is sufficient to be budget-aware during training. This ablation randomly selects $B$ actions from $\pi$ trained by BCOL and the rest from the estimated behavior policy. The setting is closest to BCOL, but the lack of planning on how to spend the budget during the testing still hurts the performance. The results on SAC-style and TD3-style implementations are consistent. This ablation shows that the effectiveness of BCOL relies on the collaboration of all three parts of our core idea: budgeting, planning on budgeting during the training, and planning on budgeting during the testing.

## 5 Related Work

Most recently proposed offline reinforcement methods rely on some mechanism to force the learned counterfactual decision policy to stay close to the data support. One approach to this end is regularizing the policy by its divergence with data, either by parameterization [12, 15], constraints and projection [29, 32, 44], divergence regularization [10, 11, 25], or implicit regularization by weighted regression [10, 37, 39, 40, 50]. The other similar idea, which is often applied together, is regularizing the value estimate of policy in an actor-critic or Q learning architecture [5, 7, 8, 22, 26, 28, 33, 34, 51]. Similar regularization can also be realized by the ensemble of $Q$ functions [1, 2, 14].

Some recent batch RL methods focus on more adaptive policy, similarly to this paper, but conditioning on history [16] or confidence level [19]. A similar idea to us of making a few key decisions in one trajectory was studied [18, 46], but they focus on stitching the logged trajectories in data.

In contrast to batch RL, imitation learning often does not involve dynamic programming on extrapolated actions. It has been used as a regularization in online RL [31, 38] and offline RL [11, 37]. Recently imitation learning methods using transformers and conditional training also achieves good performance on offline RL benchmarks [4, 56]. One-step RL [3] can also be viewed as an extension

of imitation learning with one-step policy extraction [49]. Although utilizing the behavior policy as well, Our key idea is different from these as we train a policy with awareness of behavior policy, rather than regularized by behavior cloning or imitation learning loss.

## 6  Discussions

The form of budget considered in this work is counting the different decision distributions. A natural alternative is divergences between $\pi$ and $\mu$. This provides a soft count of counterfactual decisions. However, it requires good calibration of both distributions and is more suitable to the discrete-action settings. We leave this as future work. On the theory side, a natural question about the value of budgeting counterfactuals is how the benefit of less counterfactual decisions is reflected in the theoretical properties of offline RL. We leave the investigation on this for future theoretical work.

In conclusion, this paper studies offline reinforcement learning which aims to learn a good counterfactual decision policy from a fixed dataset. We propose a novel idea of budgeting the number of counterfactual decisions and solving the allocation problem by dynamic programming. We provide strong empirical performance over offline RL benchmarks and optimality of the fixed point solution as theoretical justification.

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

# A  Proof of Theorem 2

*Proof.* We first prove that $\mathcal{T}_{\text{CB}}$ is a $\gamma$-contraction with respect to $\|\cdot\|_\infty$. For any function $Q_1, Q_2$ in the space, and any $s \in \mathcal{S}, b > 0$:

$$|V_{Q_1}(s,b) - V_{Q_2}(s,b)| \tag{10}$$

$$= \left| \max\{\max_a Q_1(s,b-1,a), \mathbf{E}_{a\sim\mu}Q_1(s,b,a)\} - \max\{\max_a Q_2(s,b-1,a), \mathbf{E}_{a\sim\mu}Q_2(s,b,a)\} \right| \tag{11}$$

$$\leq \max\{\max_a |Q_1(s,b-1,a) - Q_2(s,b-1,a)|, \mathbf{E}_{a\sim\mu}|Q_1(s,b,a) - Q_2(s,b,a)|\} \tag{12}$$

$$\leq \max\{\|Q_1 - Q_2\|_\infty, \|Q_1 - Q_2\|_\infty\} \tag{13}$$

$$= \|Q_1 - Q_2\|_\infty \tag{14}$$

The first line is from the definition of $V_Q$. The second line follows from the fact that $|\max f(x) - \max g(x)| \leq \max |f(x) - g(x)|$. The rest follows from the definition of infinity norm between Q functions. For $b = 0$, it is straightforward that $|V_{Q_1}(s,b) - V_{Q_2}(s,b)| \leq \|Q_1 - Q_2\|_\infty$ as well. So $\|V_{Q_1} - V_{Q_2}\|_\infty \leq \|Q_1 - Q_2\|_\infty$ Now we have that

$$\|\mathcal{T}_{\text{CB}}Q_1 - \mathcal{T}_{\text{CB}}Q_1\|_\infty = \|r + \gamma\,\mathbb{E}[V_{Q_1}] - r - \gamma\,\mathbb{E}[V_{Q_2}]\|_\infty \tag{15}$$

$$\leq \gamma\|Q_1 - Q_2\|_\infty \tag{16}$$

We finished the proof of $\mathcal{T}_{\text{CB}}$ is a $\gamma$-contraction operator. Thus there exists a unique fixed point of the operator $\mathcal{T}_{\text{CB}}$. Now we prove the second part of the theorem that $Q^\star = \mathcal{T}_{\text{CB}}Q^\star$.

First, we define

$$V^\star(s,b) := \max_\pi \mathbb{E}\left[\sum_{t=0}^\infty \gamma^t r_t \mid s_0 = 0, b_0 = b, \pi\right], \ \ s.t. \ b_t \geq 0 \tag{17}$$

In order to show $Q^\star = \mathcal{T}_{\text{CB}}Q^\star$, we need to prove the following two equations:

$$Q^\star(s,b,a) = r(s,a) + \gamma\,\mathbb{E}_{s'}[V^\star(s',b)] \tag{18}$$

$$V^\star(s,b) = \begin{cases} \max\{\max_a Q^\star(s,b-1,a), \mathbb{E}_{a\sim\mu}Q^\star(s,b,a)\} & b > 0 \\ \mathbb{E}_{a\sim\mu}Q^\star(s,b,a) & b = 0 \end{cases} \tag{19}$$

To prove Equation [18], we write $Q^\star(s,b,a)$ as:

$$Q^\star(s,b,a) = \max_{\pi\ s.t.b_t\geq 0} r(s,a) + \mathbb{E}\left[\sum_{t=1}^\infty \gamma^t r_t \mid s_0 = s, a_0 = a, b_1 = b, \pi\right] \tag{20}$$

$$= \max_{\pi\ s.t.b_t\geq 0} r(s,a) + \gamma\,\mathbb{E}_{s'}\left[\mathbb{E}\left[\sum_{t=1}^\infty \gamma^{t-1} r_t \mid s_1 = s', b_1 = b, \pi\right]\right] \tag{21}$$

$$= r(s,a) + \gamma\,\mathbb{E}_{s'}\left[\max_{\pi\ s.t.b_t\geq 0} \mathbb{E}\left[\sum_{t=1}^\infty \gamma^{t-1} r_t \mid s_1 = s', b_1 = b, \pi\right]\right] \tag{22}$$

$$= r(s,a) + \gamma\,\mathbb{E}_{s'}\left[V^\star(s',b)\right] \tag{23}$$

The first line holds by the linearity of expectation. The second line follows from the definition of conditional expectation. The third line follows from the fact that given $s, a, s'$ does not depend on $\pi$. The fourth line follows from the definition of $V^\star$ and the change of index $t' = t + 1$.

Next, we prove Equation 19. When $b = 0$, it is obvious that $Q^\star = Q^\mu$ and $V^\star = V^\mu$. So $V^\star(s,0) = \mathbb{E}_{a\sim\mu}[Q^\star(s,0,a)]$. Now we consider the case of $b > 0$.

When $b > 0$, we have that

$$V^\star(s, b) = \max_{\pi \ s.t. b_t \geq 0} \mathbb{E}\left[\sum_{t=0}^{\infty} \gamma^t r_t \mid s_0 = s, b_0 = b, \pi\right] \tag{24}$$

$$\leq \max_{\pi_0} \max_{\pi \ s.t. b_t \geq 0} \mathbb{E}_{a \sim \pi(\cdot|s,b)}\left[\mathbb{E}\left[\sum_{t=0}^{\infty} \gamma^t r_t \mid s_0 = 0, a_0 = a, b_0 = b, \pi\right]\right] \tag{25}$$

$$= \max_{\pi_0} \mathbb{E}_{a \sim \pi(\cdot|s,b)}\left[\max_{\pi \ s.t. b_t \geq 0} \mathbb{E}\left[\sum_{t=0}^{\infty} \gamma^t r_t \mid s_0 = 0, a_0 = a, b_0 = b, \pi\right]\right] \tag{26}$$

$$= \max_{\pi_0} \mathbb{E}_{a \sim \pi(\cdot|s,b)}\left[\max_{\pi \ s.t. b_t \geq 0} \mathbb{E}\left[\sum_{t=0}^{\infty} \gamma^t r_t \mid s_0 = 0, a_0 = a, b_1 = b - I\{\pi_0(\cdot|s,b) = \mu(\cdot|s)\}, \pi\right]\right] \tag{27}$$

$$= \max_{\pi_0} \mathbb{E}_{a \sim \pi(\cdot|s,b)} Q^\star(s, b - I\{\pi_0(\cdot|s,b) = \mu(\cdot|s)\}, a) \tag{28}$$

$$= \max\{\max_a Q^\star(s, b - 1, a), \mathbb{E}_{a \sim \mu} Q^\star(s, b, a)\} \tag{29}$$

Next, we need to prove that

$$\max_{\pi \ s.t. b_t \geq 0} \mathbb{E}\left[\sum_{t=0}^{\infty} \gamma^t r_t \mid s_0 = s, b_0 = b, \pi\right] \tag{30}$$

$$= \max_{\pi_0} \max_{\pi \ s.t. b_t \geq 0} \mathbb{E}_{a \sim \pi(\cdot|s,b)}\left[\mathbb{E}\left[\sum_{t=0}^{\infty} \gamma^t r_t \mid s_0 = 0, a_0 = a, b_0 = b, \pi\right]\right] \tag{31}$$

Let

$$\pi^\star = \operatorname*{argmax}_{\pi \ s.t. b_t \geq 0} \mathbb{E}\left[\sum_{t=0}^{\infty} \gamma^t r_t \mid s_0 = s, b_0 = b, \pi\right] \tag{32}$$

$$\pi_0^\star = \operatorname*{argmax}_{\pi_0} \max_{\pi \ s.t. b_t \geq 0} \mathbb{E}_{a \sim \pi(\cdot|s,b)}\left[\mathbb{E}\left[\sum_{t=0}^{\infty} \gamma^t r_t \mid s_0 = 0, a_0 = a, b_0 = b, \pi\right]\right] \tag{33}$$

If

$$\max_{\pi \ s.t. b_t \geq 0} \mathbb{E}\left[\sum_{t=0}^{\infty} \gamma^t r_t \mid s_0 = s, b_0 = b, \pi\right] \tag{34}$$

$$< \max_{\pi_0} \max_{\pi \ s.t. b_t \geq 0} \mathbb{E}_{a \sim \pi(\cdot|s,b)}\left[\mathbb{E}\left[\sum_{t=0}^{\infty} \gamma^t r_t \mid s_0 = 0, a_0 = a, b_0 = b, \pi\right]\right], \tag{35}$$

then it must be the case that $\pi^\star(\cdot|s, b)$ is not equal to $\pi_0^\star(\cdot|s, b)$. Define

$$\bar{\pi}(\cdot|s_t, b_t) = \begin{cases} \pi_0^\star(\cdot|s_t, b_t) & s_t = s, b_t = b \\ \pi^\star(\cdot|s_t, b_t) & \text{o.w.} \end{cases} \tag{36}$$

Then we have

$$\max_{\pi \ s.t. b_t \geq 0} \mathbb{E}\left[\sum_{t=0}^{\infty} \gamma^t r_t \mid s_0 = s, b_0 = b, \pi\right] \tag{37}$$

$$< \max_{\pi_0} \max_{\pi \ s.t. b_t \geq 0} \mathbb{E}_{a \sim \pi(\cdot|s,b)}\left[\mathbb{E}\left[\sum_{t=0}^{\infty} \gamma^t r_t \mid s_0 = 0, a_0 = a, b_0 = b, \pi\right]\right] \tag{38}$$

$$< \mathbb{E}_{a \sim \pi_0^\star(\cdot|s,b)}\left[\mathbb{E}\left[\sum_{t=0}^{\infty} \gamma^t r_t \mid s_0 = 0, a_0 = a, b_0 = b, \pi^\star\right]\right] \tag{39}$$

$$< \mathbb{E}_{a \sim \pi_0^\star(\cdot|s,b)}\left[\mathbb{E}\left[\sum_{t=0}^{\infty} \gamma^t r_t \mid s_0 = 0, a_0 = a, b_0 = b, \bar{\pi}\right]\right] \tag{40}$$

$$= \mathbb{E}\left[\sum_{t=0}^{\infty} \gamma^t r_t \mid s_0 = s, b_0 = b, \bar{\pi}\right] \tag{41}$$

By the definition of $\pi_0^\star$ and $b > 0$, $\bar{\pi}$ satisfy the constraints $b_t \geq 0$ as well. Thus it contradicts the definition of maximum. So we finished the proof of

$$V^\star(s,b) = \max\{\max_a Q^\star(s, b-1, a), \ \mathbb{E}_{a \sim \mu} Q^\star(s,b,a)\}, \ \forall s \in \mathcal{S}, \ b > 0 \tag{42}$$

This finished the proof of the fixed point of $\mathcal{T}_{\text{CB}}$ is $Q^\star$. $\qquad\square$

# B    Experimental Details

| Hyperparameter | BCOL SAC | BCOL TD3 |
|---|---|---|
| Random seeds | $\{0, 1, 2\}$ | |
| Number of gradient steps | $1e6$ | |
| Hidden layer dimensions | $[256, 256, 256]$ | $[256, 256]$ |
| Activation function | ReLU | |
| Discounting factor ($\gamma$) | 0.99 | |
| Target network update rate | 0.005 | |
| Optimizer | Adam | |
| Actor learning rate | $3e - 4$ [1] | $3e - 4$ |
| Critic learning rate | $7e - 4$ | $3e - 4$ |
| Number of Q functions | 2 | |
| Coefficient of $\min$ Q values ($\lambda$ in [12]) | 0.75 | 1 |
| Number of actions samples ($m$) | 5 | - |
| TD3 policy noise | - | 0.2 |
| TD3 noise clip | - | 0.5 |
| TD3 policy update freq | - | 2 |
| Counterfactual budget $B$ | $\{1, 10, 50\}$ | |
| Monotonicity penalty coefficient $\omega$ | $\{0, 1, 10, 100\}$ | |

**Table 2:** Hyperparameter values and model architecture details in BCOL.

## B.1    Hyper-parameters and infrastructures of experiments

We implement BCOL with the same architecture and hyper-parameter values as baseline methods. We list all hyperparameter values and neural architecture details in Table 2. For SAC-based implementation, we follow CDC's [8] hyper-parameter values. The only exception is that we use $2$ $Q$ functions instead of 4, and we sample $5$ actions from actor ($m$) instead of 15. This is to save the amount of computation with a small amount of performance decrease for our algorithm. The baseline method (offline SAC) uses the same hyperparameter values as CDC. For TD3-based implementation, we follows TD3 [13] and TD3+BC's [11] hyper-parameter values.

To generate the final results as well as learning curves, we run all algorithms with $1,000,000$ policy gradient steps, following prior work. We evaluate algorithms every $5,000$ step before the $900,000$th gradient step, and every $1,000$ step after the $900,000$th gradient step. For each evaluation, we run 10 test episodes in the environment.

Table 3 describes the hardware infrastructure and software libraries we used to run the experiment in this paper. We will release the code to reproduce the experiment upon publication.

## B.2    Important experimental setup for AntMaze

Among prior offline RL work, IQL [23] and CQL [26] outperform other algorithms with a large margin in AntMaze domains. We notice that these methods also use some implementation tricks that are different from other work, in AntMaze domains. In order to clarify and eliminate all possible orthogonal factors, we list these implementation details below and implement our algorithm BCOL and two baselines (CDC and TD3+BC) with these implementations.

IQL uses a cosine learning rate schedule for the actor learning rate and a state-independent standard deviation in the policy network. Most prior work such as TD3+BC and CDC does not use these implementations. According to Appendix B in the IQL paper, CQL and IQL also subtract 1 from rewards for the AntMaze datasets. We apply these tricks for BCOL and two baselines (CDC and TD3+BC) that we re-implement, and report the AntMaze results in Table 1.

---

[1]We use learning rate $3e - 4$ for Gym domains and $1e - 4$ for AntMaze domains. Following the suggestion from CDC authors, we found out that CDC and offline SAC perform better with a smaller learning rate in AntMaze domains. Thus we set this learning rate for our algorithm BCOL SAC and baselines (CDC, offline SAC).

| Machine Type | AWS EC2 g4dn.2xlarge |
|---|---|
| GPU | Tesla T4 |
| CPU | Intel Xeon 2.5GHz |
| CUDA version | 11.0 |
| NVIDIA Driver | 450.142.00 |
| PyTorch version | 1.12.1 |
| Gym version | 19.0 |
| Python version | 3.8.13 |
| NumPy version | 1.21.5 |
| D4RL datasets version | v2 |

**Table 3:** Hardware infrastructure and software libraries used in this paper

## B.3 Budget consumption analysis

One might wonder that if BCOL spend all the budget in the beginning of trajectories. In fact, our experiment shows there are still budgets left at the end of the episode during the test. In Table 4 we present the total BCOL spent in the test (plus minus the standard deviation), averaged over seeds and last 10 policies, for AntMaze tasks. This result shows that the algorithm will not always be forced by Line 2 in `Select` in later steps. In contrast, it will plan on how to spend the budget by Line 1 in `Select`.

| Task | Average budget spent ($B = 50$) |
|---|---|
| Antmaze-umaze | $41.00 \pm 0.91$ |
| Antmaze-umaze-diverse | $20.02 \pm 6.44$ |
| Antmaze-medium-play | $45.94 \pm 1.13$ |
| Antmaze-medium-diverse | $46.04 \pm 0.92$ |
| Antmaze-large-play | $43.33 \pm 1.81$ |
| Antmaze-large-diverse | $43.75 \pm 1.77$ |

**Table 4:** Budget Consumption

## B.4 Comparison to additional baselines

Here we include the comparison against more baselines. These baselines are not discussed in the main paper either because they lack of result in the more challenging Antmaze tasks, or due to their relative inferior performance compared to the more recent offline RL algorithms. The numbers are from the original paper if available (ATAC, RIQL, CRR+, CQL+), or from other paper's report (MOPO, MoREL and COMBO is from [42] and MuZero is from [20]).

| Algorithm | Mujoco Total | Antmaze Total | Total |
|---|---|---|---|
| BCOL (SAC) | 746.0 | **396.0** | **1142.0** |
| MoREL [21] | 656.5 | 0 | 656.5 |
| MOPO [55] | 379.3 | 0 | 379.3 |
| COMBO [54] | 738.3 | 137.6 | 875.9 |
| MuZero [45] | 140.2 | 0 | 140.2 |
| RIQL [10] | 759.1 | - | - |
| ATAC [5] | **792.4** | - | - |
| CRR+ [20] | 703.9 | 41.9 | 745.8 |
| CQL+ [20] | 717.9 | 89.0 | 806.9 |

**Table 5:** Additional baselines

### B.5 Complete experimental results

In this section, we include experimental details that are deferred here due to the space limitations of the main paper. First, we report the average and standard deviation (not included in Table 1) of all algorithms that we implemented in Table 6. The standard deviation is calculated across the 300 evaluation episodes for each algorithm (3 random seeds, 10 evaluation steps per random seed, and 10 episodes per evaluation.).

| Task Name | TD3+BC | BCOL (TD3) | CDC | BCOL (SAC) |
|---|---|---|---|---|
| halfcheetah-medium-v2 | $48.4 \pm 0.3$ | $45.0 \pm 0.4$ | $62.5 \pm 1.2$ | $50.1 \pm 0.3$ |
| hopper-medium-v2 | $59.4 \pm 4.6$ | $85.8 \pm 10.2$ | $84.9 \pm 8.4$ | $83.2 \pm 7.5$ |
| walker2d-medium-v2 | $84.6 \pm 1.6$ | $76.5 \pm 5.8$ | $70.7 \pm 16.6$ | $84.1 \pm 1.5$ |
| halfcheetah-medium-replay-v2 | $44.5 \pm 0.5$ | $41.1 \pm 0.9$ | $52.3 \pm 1.7$ | $46.2 \pm 1.0$ |
| hopper-medium-replay-v2 | $50.1 \pm 22.2$ | $81.8 \pm 19.2$ | $87.4 \pm 13.2$ | $99.8 \pm 1.4$ |
| walker2d-medium-replay-v2 | $80.2 \pm 7.6$ | $49.6 \pm 15.2$ | $87.8 \pm 5.4$ | $86.0 \pm 3.3$ |
| halfcheetah-medium-expert-v2 | $91.5 \pm 5.2$ | $89.0 \pm 3.8$ | $66.3 \pm 7.0$ | $86.9 \pm 5.7$ |
| hopper-medium-expert-v2 | $100.5 \pm 11.8$ | $106.9 \pm 7.2$ | $83.2 \pm 16.3$ | $99.0 \pm 11.6$ |
| walker2d-medium-expert-v2 | $110.1 \pm 0.5$ | $108.6 \pm 0.4$ | $103.9 \pm 7.5$ | $110.8 \pm 0.4$ |
| antmaze-umaze-v2 | $96.3 \pm 5.5$ | $93.3 \pm 8.3$ | $93.7 \pm 8.7$ | $90.3 \pm 8.4$ |
| antmaze-umaze-diverse-v2 | $71.7 \pm 15.3$ | $68.0 \pm 15.8$ | $57.3 \pm 26.2$ | $90.0 \pm 8.9$ |
| antmaze-medium-play-v2 | $1.7 \pm 3.7$ | $12.3 \pm 18.7$ | $59.5 \pm 15.6$ | $70.0 \pm 12.6$ |
| antmaze-medium-diverse-v2 | $1.3 \pm 4.3$ | $14.0 \pm 13.1$ | $64.7 \pm 14.3$ | $72.3 \pm 17.5$ |
| antmaze-large-play-v2 | $0.0 \pm 0.0$ | $0.0 \pm 0.0$ | $33.0 \pm 12.7$ | $35.7 \pm 23.6$ |
| antmaze-large-diverse-v2 | $0.3 \pm 1.8$ | $0.0 \pm 0.0$ | $25.3 \pm 15.4$ | $37.7 \pm 18.2$ |

**Table 6:** Average normalized score and standard deviation on D4RL tasks.

Second, we include the learning curves of these 4 algorithms in Fig. 4 to present the details of training. Each data point in the learning curve represents one evaluation step and is averaged over 30 test episodes (3 random seeds and 10 test episodes per evaluation). The shadow region represents the standard deviation over 30 test scores. For a better visibility, we smooth the learning curves by moving averages with a window size of 10.

Third, we include the performance of BCOL with all searched hyper-parameter values on each task in Table 7 (for BCOL SAC) and Table 8 (for BCOL TD3).

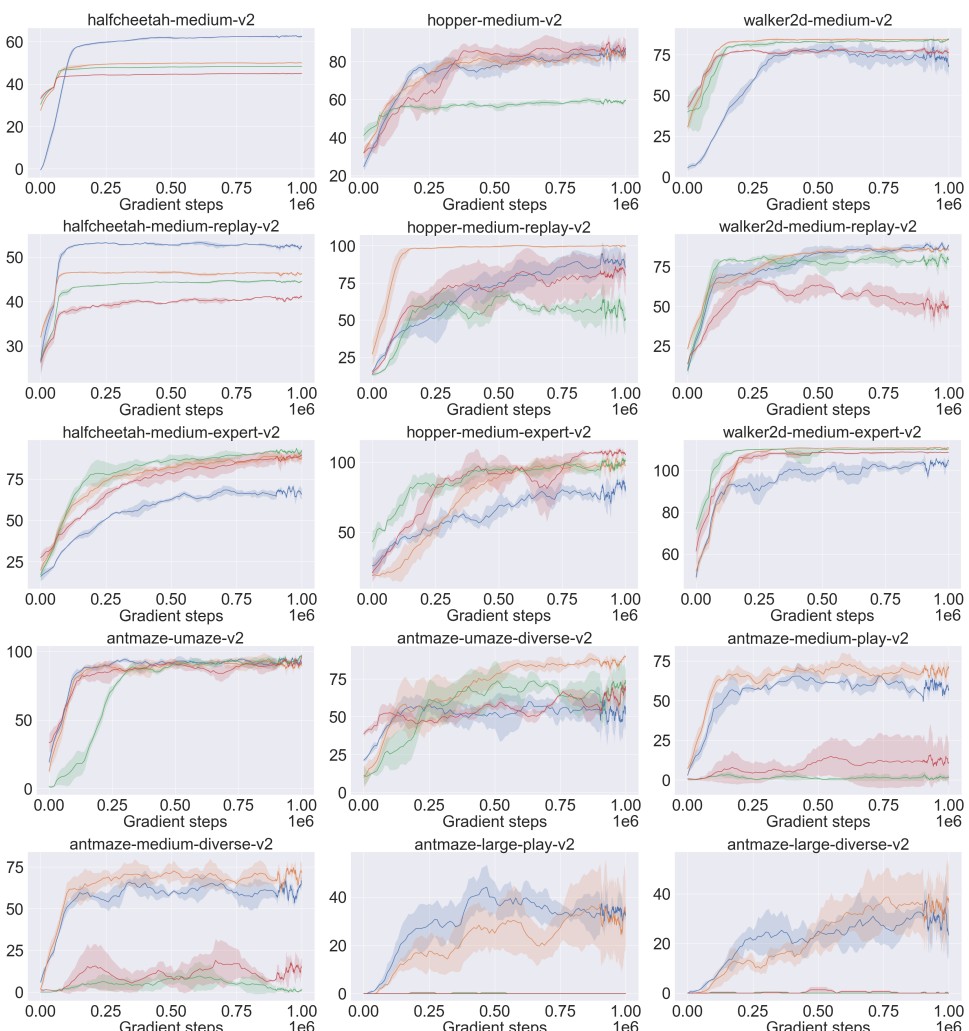

**Figure 4:** Learning curves for CDC, BCOL (SAC), TD3+BC, and BCOL (TD3) in D4RL tasks.

| Task Name | B = 1 | | | | B = 10 | | | | B = 50 | | | |
|---|---|---|---|---|---|---|---|---|---|---|---|---|
| | $\omega = 0$ | $\omega = 1$ | $\omega = 10$ | $\omega = 100$ | $\omega = 0$ | $\omega = 1$ | $\omega = 10$ | $\omega = 100$ | $\omega = 0$ | $\omega = 1$ | $\omega = 10$ | $\omega = 100$ |
| halfcheetah-medium-v2 | $42.7 \pm 0.4$ | $42.7 \pm 0.3$ | $42.8 \pm 0.4$ | $42.8 \pm 0.5$ | $43.7 \pm 0.4$ | $43.8 \pm 0.3$ | $43.5 \pm 0.6$ | $43.7 \pm 0.4$ | $44.3 \pm 1.2$ | $45.3 \pm 0.3$ | $45.0 \pm 0.4$ | $42.5 \pm 0.4$ |
| hopper-medium-v2 | $51.6 \pm 5.5$ | $55.2 \pm 7.1$ | $52.8 \pm 7.2$ | $55.5 \pm 6.3$ | $53.7 \pm 5.7$ | $59.2 \pm 5.7$ | $57.7 \pm 5.0$ | $78.1 \pm 15.4$ | $7.7 \pm 10.0$ | $59.9 \pm 28.8$ | $85.8 \pm 10.2$ | $61.1 \pm 10.2$ |
| walker2d-medium-v2 | $72.1 \pm 4.9$ | $70.0 \pm 6.7$ | $71.4 \pm 5.9$ | $71.6 \pm 5.8$ | $76.8 \pm 4.0$ | $77.6 \pm 3.9$ | $77.5 \pm 4.2$ | $79.1 \pm 3.1$ | $0.2 \pm 1.6$ | $51.0 \pm 30.7$ | $76.5 \pm 5.8$ | $70.2 \pm 4.9$ |
| halfcheetah-medium-replay-v2 | $36.5 \pm 3.2$ | $36.5 \pm 2.8$ | $36.5 \pm 2.0$ | $36.6 \pm 2.1$ | $40.1 \pm 1.0$ | $40.4 \pm 0.7$ | $39.9 \pm 1.3$ | $40.0 \pm 1.0$ | $41.3 \pm 1.5$ | $42.1 \pm 0.8$ | $41.1 \pm 0.9$ | $34.2 \pm 3.6$ |
| hopper-medium-replay-v2 | $28.9 \pm 9.1$ | $29.5 \pm 10.7$ | $29.0 \pm 9.1$ | $28.3 \pm 8.9$ | $40.8 \pm 14.1$ | $46.6 \pm 13.8$ | $51.3 \pm 12.7$ | $57.4 \pm 19.4$ | $43.7 \pm 8.5$ | $62.1 \pm 19.0$ | $81.8 \pm 19.2$ | $25.0 \pm 8.1$ |
| walker2d-medium-replay-v2 | $26.4 \pm 11.3$ | $25.3 \pm 10.7$ | $25.7 \pm 9.0$ | $25.2 \pm 10.9$ | $57.3 \pm 8.9$ | $56.5 \pm 10.9$ | $56.3 \pm 12.2$ | $58.7 \pm 10.9$ | $57.5 \pm 16.3$ | $72.7 \pm 7.4$ | $49.6 \pm 15.2$ | $17.6 \pm 8.1$ |
| halfcheetah-medium-expert-v2 | $69.7 \pm 7.9$ | $70.1 \pm 8.2$ | $72.7 \pm 7.1$ | $72.9 \pm 8.1$ | $80.6 \pm 6.2$ | $85.6 \pm 7.0$ | $90.3 \pm 2.6$ | $91.2 \pm 1.5$ | $60.2 \pm 14.5$ | $84.2 \pm 5.6$ | $89.0 \pm 3.8$ | $84.2 \pm 4.9$ |
| hopper-medium-expert-v2 | $77.4 \pm 18.4$ | $83.6 \pm 20.5$ | $90.8 \pm 18.0$ | $78.3 \pm 21.0$ | $33.0 \pm 26.9$ | $82.3 \pm 14.2$ | $102.9 \pm 14.4$ | $106.2 \pm 5.8$ | $1.9 \pm 1.2$ | $31.2 \pm 26.8$ | $106.9 \pm 7.2$ | $76.2 \pm 45.6$ |
| walker2d-medium-expert-v2 | $106.7 \pm 3.5$ | $106.4 \pm 4.6$ | $105.7 \pm 6.1$ | $106.8 \pm 3.9$ | $87.4 \pm 18.4$ | $107.0 \pm 5.5$ | $108.7 \pm 0.3$ | $108.6 \pm 0.4$ | $-0.2 \pm 0.0$ | $87.8 \pm 26.3$ | $108.6 \pm 0.4$ | $93.8 \pm 16.5$ |
| antmaze-umaze-v2 | $44.7 \pm 12.6$ | $45.3 \pm 12.3$ | $45.3 \pm 12.3$ | $45.3 \pm 12.6$ | $58.3 \pm 15.1$ | $55.7 \pm 15.4$ | $41.3 \pm 13.1$ | $41.3 \pm 13.8$ | $93.3 \pm 8.3$ | $86.3 \pm 12.5$ | $50.3 \pm 13.3$ | $49.7 \pm 13.0$ |
| antmaze-umaze-diverse-v2 | $48.3 \pm 15.9$ | $46.0 \pm 16.9$ | $46.0 \pm 16.9$ | $46.0 \pm 16.9$ | $48.0 \pm 18.3$ | $44.3 \pm 17.1$ | $44.7 \pm 18.0$ | $45.0 \pm 18.8$ | $68.0 \pm 15.8$ | $51.0 \pm 19.2$ | $53.3 \pm 17.8$ | $45.3 \pm 15.2$ |
| antmaze-medium-play-v2 | $0.7 \pm 2.5$ | $0.3 \pm 1.8$ | $0.7 \pm 2.5$ | $0.7 \pm 2.5$ | $0.0 \pm 0.0$ | $0.0 \pm 0.0$ | $0.0 \pm 0.0$ | $0.0 \pm 0.0$ | $12.3 \pm 18.7$ | $0.0 \pm 0.0$ | $0.0 \pm 0.0$ | $0.0 \pm 0.0$ |
| antmaze-medium-diverse-v2 | $0.0 \pm 0.0$ | $0.0 \pm 0.0$ | $0.0 \pm 0.0$ | $0.0 \pm 0.0$ | $0.0 \pm 0.0$ | $0.0 \pm 0.0$ | $0.0 \pm 0.0$ | $0.0 \pm 0.0$ | $14.0 \pm 13.1$ | $0.0 \pm 0.0$ | $0.0 \pm 0.0$ | $0.0 \pm 0.0$ |
| antmaze-large-play-v2 | $0.0 \pm 0.0$ | $0.0 \pm 0.0$ | $0.0 \pm 0.0$ | $0.0 \pm 0.0$ | $0.0 \pm 0.0$ | $0.0 \pm 0.0$ | $0.0 \pm 0.0$ | $0.0 \pm 0.0$ | $0.0 \pm 0.0$ | $0.0 \pm 0.0$ | $0.0 \pm 0.0$ | $0.0 \pm 0.0$ |
| antmaze-large-diverse-v2 | $0.0 \pm 0.0$ | $0.0 \pm 0.0$ | $0.0 \pm 0.0$ | $0.0 \pm 0.0$ | $0.0 \pm 0.0$ | $0.0 \pm 0.0$ | $0.0 \pm 0.0$ | $0.0 \pm 0.0$ | $0.0 \pm 0.0$ | $0.0 \pm 0.0$ | $0.0 \pm 0.0$ | $0.0 \pm 0.0$ |

**Table 7:** Average normalize score and standard deviation of hyper-parameter ablation on BCOL (TD3).

| Task Name | B = 1 | | | | B = 10 | | | | B = 50 | | | |
|---|---|---|---|---|---|---|---|---|---|---|---|---|
| | $\omega = 0$ | $\omega = 1$ | $\omega = 10$ | $\omega = 100$ | $\omega = 0$ | $\omega = 1$ | $\omega = 10$ | $\omega = 100$ | $\omega = 0$ | $\omega = 1$ | $\omega = 10$ | $\omega = 100$ |
| halfcheetah-medium-v2 | $50.7 \pm 0.3$ | $50.7 \pm 0.4$ | $50.4 \pm 0.2$ | $49.7 \pm 0.2$ | $52.6 \pm 0.8$ | $52.3 \pm 0.4$ | $50.1 \pm 0.3$ | $48.2 \pm 0.4$ | $53.1 \pm 1.5$ | $53.7 \pm 0.3$ | $46.9 \pm 0.4$ | $39.7 \pm 1.0$ |
| hopper-medium-v2 | $73.3 \pm 8.6$ | $72.2 \pm 8.4$ | $73.5 \pm 8.0$ | $73.9 \pm 9.8$ | $63.8 \pm 17.0$ | $71.7 \pm 8.0$ | $83.2 \pm 7.5$ | $61.9 \pm 7.8$ | $4.9 \pm 6.2$ | $71.0 \pm 17.9$ | $58.3 \pm 5.9$ | $50.9 \pm 10.4$ |
| walker2d-medium-v2 | $84.5 \pm 2.5$ | $84.9 \pm 3.0$ | $85.1 \pm 2.2$ | $84.6 \pm 0.4$ | $33.9 \pm 34.7$ | $80.9 \pm 7.8$ | $84.1 \pm 1.5$ | $83.2 \pm 1.8$ | $42.7 \pm 24.2$ | $52.2 \pm 30.5$ | $81.4 \pm 3.1$ | $54.9 \pm 12.6$ |
| halfcheetah-medium-replay-v2 | $43.7 \pm 1.8$ | $43.1 \pm 2.5$ | $43.8 \pm 1.2$ | $44.0 \pm 1.8$ | $45.6 \pm 1.7$ | $46.1 \pm 1.4$ | $46.2 \pm 1.0$ | $44.2 \pm 1.2$ | $21.7 \pm 16.6$ | $48.1 \pm 1.4$ | $41.0 \pm 1.6$ | $15.4 \pm 6.7$ |
| hopper-medium-replay-v2 | $91.9 \pm 9.7$ | $86.0 \pm 11.0$ | $91.8 \pm 7.2$ | $93.2 \pm 8.0$ | $96.1 \pm 6.5$ | $92.9 \pm 8.7$ | $99.8 \pm 1.4$ | $88.6 \pm 12.6$ | $11.6 \pm 13.1$ | $93.6 \pm 9.9$ | $79.5 \pm 16.9$ | $4.0 \pm 6.0$ |
| walker2d-medium-replay-v2 | $75.2 \pm 8.6$ | $76.5 \pm 9.8$ | $79.4 \pm 5.5$ | $81.1 \pm 5.7$ | $66.9 \pm 12.9$ | $74.4 \pm 11.3$ | $86.0 \pm 3.3$ | $82.1 \pm 5.1$ | $9.4 \pm 9.5$ | $51.9 \pm 14.0$ | $30.9 \pm 26.4$ | $5.4 \pm 6.9$ |
| halfcheetah-medium-expert-v2 | $79.5 \pm 5.3$ | $81.1 \pm 5.9$ | $84.6 \pm 6.5$ | $85.6 \pm 5.5$ | $79.7 \pm 7.6$ | $80.5 \pm 7.2$ | $86.9 \pm 5.7$ | $83.3 \pm 5.7$ | $65.4 \pm 9.6$ | $73.8 \pm 9.0$ | $83.6 \pm 5.9$ | $41.7 \pm 3.2$ |
| hopper-medium-expert-v2 | $76.3 \pm 18.8$ | $70.2 \pm 19.4$ | $88.2 \pm 15.2$ | $90.6 \pm 12.0$ | $16.2 \pm 17.9$ | $68.1 \pm 17.0$ | $99.0 \pm 11.6$ | $92.9 \pm 13.0$ | $4.2 \pm 2.8$ | $38.3 \pm 15.9$ | $99.0 \pm 10.6$ | $50.1 \pm 10.5$ |
| walker2d-medium-expert-v2 | $110.6 \pm 2.3$ | $110.7 \pm 0.7$ | $110.8 \pm 0.4$ | $109.5 \pm 2.7$ | $42.0 \pm 43.4$ | $110.3 \pm 1.2$ | $110.8 \pm 0.4$ | $109.7 \pm 0.4$ | $64.7 \pm 31.7$ | $93.0 \pm 18.3$ | $109.7 \pm 0.7$ | $71.1 \pm 27.6$ |
| antmaze-umaze-v2 | $31.0 \pm 34.2$ | $73.0 \pm 16.2$ | $72.0 \pm 13.8$ | $65.3 \pm 20.0$ | $91.0 \pm 7.9$ | $91.7 \pm 8.2$ | $85.3 \pm 9.6$ | $75.7 \pm 13.1$ | $90.3 \pm 8.4$ | $91.7 \pm 8.6$ | $64.0 \pm 14.7$ | $83.3 \pm 14.7$ |
| antmaze-umaze-diverse-v2 | $38.0 \pm 18.9$ | $54.0 \pm 16.9$ | $54.7 \pm 16.7$ | $62.0 \pm 14.5$ | $4.3 \pm 8.4$ | $55.0 \pm 14.3$ | $60.7 \pm 14.8$ | $56.3 \pm 18.2$ | $90.0 \pm 8.9$ | $54.0 \pm 15.8$ | $33.3 \pm 20.1$ | $49.0 \pm 29.1$ |
| antmaze-medium-play-v2 | $0.0 \pm 0.0$ | $0.0 \pm 0.0$ | $0.0 \pm 0.0$ | $0.3 \pm 1.8$ | $14.0 \pm 19.3$ | $7.7 \pm 12.3$ | $0.0 \pm 0.0$ | $0.0 \pm 0.0$ | $70.0 \pm 12.6$ | $26.7 \pm 19.9$ | $0.0 \pm 0.0$ | $0.0 \pm 0.0$ |
| antmaze-medium-diverse-v2 | $0.0 \pm 0.0$ | $0.0 \pm 0.0$ | $0.3 \pm 1.8$ | $0.0 \pm 0.0$ | $4.0 \pm 7.6$ | $9.7 \pm 8.4$ | $0.0 \pm 0.0$ | $0.0 \pm 0.0$ | $72.3 \pm 17.5$ | $23.7 \pm 14.3$ | $0.0 \pm 0.0$ | $0.3 \pm 1.8$ |
| antmaze-large-play-v2 | $0.0 \pm 0.0$ | $0.0 \pm 0.0$ | $0.0 \pm 0.0$ | $0.0 \pm 0.0$ | $0.0 \pm 0.0$ | $0.0 \pm 0.0$ | $0.0 \pm 0.0$ | $0.0 \pm 0.0$ | $35.7 \pm 23.6$ | $2.0 \pm 6.0$ | $0.0 \pm 0.0$ | $0.0 \pm 0.0$ |
| antmaze-large-diverse-v2 | $0.0 \pm 0.0$ | $0.0 \pm 0.0$ | $0.0 \pm 0.0$ | $0.0 \pm 0.0$ | $0.0 \pm 0.0$ | $0.0 \pm 0.0$ | $0.0 \pm 0.0$ | $0.0 \pm 0.0$ | $37.7 \pm 18.2$ | $0.0 \pm 0.0$ | $0.0 \pm 0.0$ | $0.0 \pm 0.0$ |

**Table 8:** Average normalize score and standard deviation of hyper-parameter ablation on BCOL (SAC).

