# OpenReview forum: "Budgeting Counterfactual for Offline RL"
_NeurIPS.cc/2023/Conference — NeurIPS 2023 poster_

### Official Review · Reviewer_bnLz · 2023-07-03

**Soundness:** 2 fair
**Presentation:** 2 fair
**Contribution:** 2 fair
**Rating:** 5
**Confidence:** 4

**Summary:**

This paper proposes a novel offline RL algorithm BCOL that builds on the idea of limiting the numbers of counterfactual decisions. Instead of enforcing policy or value regularization, BCOL follows the decisions of the behavioral policy in the majority of the states, and only makes counterfactual decisions for a limited number of times. The strategy to spend the fixed amount of opportunities for counterfactual decisions is learned through a dynamic programming algorithm. Experimental results using two different implementations of the proposed algorithm on a wide range of offline RL tasks are presented.

**Strengths:**

It is great that the authors present all details about their experiments and the implementation details of their proposed algorithm.

The authors present two implementations based on different state-of-the-art RL algorithms (TD3 and SAC).

The presentation about the empirical results (figures, tables, etc.) is very clear. The ablation study on the hyper-parameters is great.


**Weaknesses:**

### Method
It is well-known in the offline RL community that the extrapolation needs to be careful. Limiting the level of extrapolation itself is not challenging, one can copy the behavioral policy, leading to zero extrapolation. The challenging thing is to find out where to extrapolate. The authors have argued for many times in the paper that assigning an upper bound for the number of counterfactual decisions can effectively constrain the level of extrapolation, leading to a balance between the gain of counterfactual decisions and the risk of extrapolation. However, this only explains how the level of extrapolation is limited but not the more important question: why the proposed BCOL algorithm can learn where to extrapolate.

On the intuitive level, the explanation is not enough and not clear. On the formal side, it would be great if the authors can provide some theoretical guarantees for the proposed algorithm so that the benefits of BCOL become more clear. In the current version of the paper, I fail to see enough support, either intuitive or theoretical, for the efficacy of the proposed algorithm.

### Algorithm
The proposed algorithm induces an extra burden for learning because the Q(s,b,a), unlike the regular state-action value function, needs to approximate the value of the budget well. This at least linearly increases the difficulty of the learning problem because for every b, Q(s,b,a) needs to be approximated well so that the proposed Counterfactual-Budgeting Bellman Operator can work well.

### Experiments
It is appreciated that the details about the experiments are presented well.

However, the performance of BCOL doesn’t seem that impressive given the increased training cost. For example,although the BCOL(SAC) has the highest total score, it is outperformed by CQL and CDC on about half of the tasks.

Moreover, the baseline methods, as diverse as they are, are dated algorithms. It may be better to include more latest baselines that also report SOTA performance in their papers (see for example, [1] and [2]) .

[1] Bhardwaj, Mohak, et al. "Adversarial model for offline reinforcement learning." arXiv preprint arXiv:2302.11048 (2023).
[2] Kang, Bingyi, et al. "Improving and Benchmarking Offline Reinforcement Learning Algorithms." arXiv preprint arXiv:2306.00972 (2023).


**Questions:**

Is the algorithm still working if there is more than one behavior policy for collecting the offline data?

The sentence “Thus, the choice of policy on each step needs to balance between the Q value gain from the current step and the potential benefit from future counterfactual decisions.” is very confusing. Could the authors please elaborate more on it?

What is the formal definition of Q(s,b,a) in Eq. (4)? It looks like the exact definition of Q(s,b,a) doesn’t matter, and the only thing that matters is the fixed point of $\mathcal{T}_{CB}$.

I understand that by definition of $\mathcal{T}_{CB}$ that are at most B backup steps taking the max operation as the maximum value. But why does this “intuitively upper bounds the amount of extrapolation that the Q function can take”?

**Limitations:**

Yes. The authors have discussed the limitations of this work and pointed out future directions for continuing research.

---

> ### Author Rebuttal · Authors · 2023-08-10
>
> Thank you for your valuable feedback. We hope you will consider increasing your score after reading our responses. Please let us know if there are more questions
>
> > “why BCOL can learn where to extrapolate"
>
> > “not enough intuitive or theoretical support”.
>
> Indeed, it is the very central question that motivates our algorithm design and we would like to clarify it and avoid misunderstanding about it. In short, the dynamic programming formula Eq (4) in BCOL is naturally derived from the constrained objective Eq (2), to find where to extrapolate under the budget. Formally, the fixed point of Eq (4) is guaranteed to be the optimal allocation of the counterfactual or extrapolation across the time steps, by Theorem 2. BCOL is the approximate dynamics programming algorithm based on Eq (4).
>
> Here we will give a more intuitive summarization of the formal discussion in paper. To solve the problem about “where to extrapolate”, we need to balance the trade-off between spending the budget now to get better action and keeping the budget for the future to get a better future value. With BCOL, we solve the trade-off by dynamic programming. Mathematically, it is described by the selection between $\max_{a’} Q(s’,a’,b-1)$ and $E_{a’ \sim \mu} Q(s’,a’,b)$ in Eq (4) in our paper. Theorem 2 says, by using such a modified Bellman backup, the fixed point is guaranteed to be the constrained optimal value function.
>
> In summary, we respectfully disagree that we do not have enough intuitive or theoretical support in the paper. We would like to refer to the description of our algorithm from reviewer wQce, who said that our algorithm is “a solution to solve the allocation problem by dynamic programming, which is also solid and theoretical justificated.”
>
> > “Computation cost”
>
> We acknowledge that BCOL will increase the amount of computation compared with SAC and TD3. In our paper, we discussed a stochastic approximation to reduce the linear cost on $B$ at the end of section 3.3. We remark that the computation is often not the main bottleneck in offline RL settings where the data is fixed (as discussed in all previous offline RL papers). It is more important to consider how to use the available data and leverage more information.
>
> > “comparison with CQL and CDC”
>
> To answer your concern, we would like to explain a bit on the difference between Mujoco and AntMaze tasks, and why we think our algorithm shows a significant improvement. Mujoco datasets in D4RL have a significant fraction of near-optimal trajectories, and are considered as easier tasks by prior work e.g. IQL and CDC. AntMaze tasks require stitching parts of suboptimal trajectories that travel between different states to find a path from the start to the goal of the maze, and are more challenging and meaningful. (More discussion in section 4 of paper.) Thus we argue that the improvements on the AntMaze tasks are more significant and valuable. On AntMaze tasks, we outperform CQL and CDC by a much larger margin.
>
> We would like to refer to our response to Reviewer 4X4f for a remark on how CQL results are reported in literature.
>
> > “latest baselines”
>
> We thank the reviewer for the pointers and we will include more baselines in an updated version. In the response to all reviewers, we compared against the two works pointed out by the reviewer, and baseline algorithms that these two works compared with. These baselines either do not report their performance, or are significantly worse than BCOL in the harder AntMaze tasks.
>
> Additionally, it is important to note Kang, Bingyi, et al. was only available online after the NeurIPS submission deadline, so it was not possible for us to consider it.
>
> > “Is the algorithm still working if there is more than one behavior policy?”
>
> Yes. Most D4RL datasets contain data from more than one behavior policy. Please see Page 5 in the D4RL paper for further details. Our results show that our method outperforms others on this benchmark. Also BCOL does not need to know the behavior policy as well, and it will fit a behavior policy as $\mu$.
>
> > “the confusing sentence”
>
> We will clarify that in the updated version, however, let us explain it here. That sentence means, if we are under the constraints that we can only take at most B counterfactual decisions, we face the problem of where to allocate them. We expect the counterfactual decisions to provide improvement on top of behavior values, but taking too many of them can result in extrapolations. At some time-step, We can take counterfactual decisions immediately, and enjoy the value improvement. We can also take action from behavior policy, and this will save the budget for the future and maybe we can use it in some more influential decisions.
>
> >  “formal definition of Q in Eq 4”
>
> Equation 4 is the definition of an operator for any Q function in the function space. Thus $Q(s,b,a)$ can be any function in the function space $\mathcal{S} \times [B] \times \mathcal{A} \to \mathbb{R} $.
>
> > “why does this intuitively upper bounds the amount of extrapolation”
>
> Good question. The bounded extrapolation is from the comparison with standard Bellman update and RL methods based on it. In a standard Bellman backup we will update the Q function by:
> $$ Q(s,a) \leftarrow r(s,a) + \gamma \max_{a’} Q(s’,a’) $$
> Practical algorithms fitting this goal keep updating the Q values with extrapolated Q value $ \max_{a’} Q(s’,a’) $ recursively (unless the argmax happens to be behavior policy). Such a recursive update is either ended by the episode end or discounted by $\gamma$. Thus there are $\frac{1}{1-\gamma}$ or $H$ (horizon) times $\max$ operator effectively on the optimization path, and that many extrapolated queries to the Q functions.
>
> In contrast, our algorithm queries at most $B$ times to the extrapolated Q values in one optimization path. As $B << \frac{1}{1-\gamma}$ and $H$, the amount of extrapolation with our Bellman backup is much smaller than with the standard Bellman operator.

---

> > ### Comment · Reviewer_bnLz · 2023-08-12
> >
> > I appreciate the author's effort in addressing my concerns. However, two of my biggest concerns are still not addressed.
> >
> > ### Theoretical Guarantee
> > The authors have repeated their motivations many times in the paper: limiting the number of times to deviate from the behavior policy can effectively improve extrapolation. The authors have accordingly defined a new operator and proved in Theorem 2 that the fixed point of the proposed operator is optimal value function of interest. However, Theorem 2, as the only theory contribution in this work, is not involved at all. More importantly, although the benefits of the proposed approach are underscored many times in words, there is no formal justification of the performance. **There is no rigorous understanding of the benefits brought by counterfactual balancing.** For example, what is the regret guarantee of BCOL? Under what condition does BCOL have an edge over other offline RL methods? These questions shadow the value of this work.
> >
> > ### Computation Cost
> > While the authors discuss very briefly that they tensorized the loop over $b$ and can use sampling over $b$ to further reduce the computation cost, a short formal analysis can show that the required number of operations for $\widehat Q_{\theta}(s,b,a)$ to uniformly reach a desired accuracy $\epsilon$ over $b$ is at least $\Omega(b)$. Moreover, the demand for high accuracy of the proposed $\widehat Q_{\theta}(s,b,a)$ should be higher than that of regular $Q(s,a)$ because the error can accumulate over backup of $b$.
> >
> > I would like to note that this is only the complexity of computation, it is possible that the the complexity of sample also increases with the introduction of $b$ into the value function, which is more challenging to address.
> >
> > Hence, I incline to keep my score for now.
> >
> > Thanks!

---

> > > ### Author Response · Authors · 2023-08-13
> > > **Response to the reviewer's further questions**
> > >
> > > We are glad to hear that our responses addressed most of the reviewer's concerns. We have addressed the remaining questions below. We encourage the reviewer to revise their score after reading our responses. We believe that we have addressed all of the reviewer's concerns.
> > >
> > > ### Theoretical Guarantee
> > >
> > > We respectfully disagree with the judgment on "lack of rigorous understanding" in this paper, from two perspectives.
> > >
> > > “The rigorous understanding of the benefits brought by counterfactual balancing (our method)” is provided by the constrained optimality (property of our solution) in Theorem 2. As for why the constrained optimality is desirable, we would like to quote the reviewer’s own initial review to explain it: “**It is well-known in the offline RL community that the extrapolation needs to be careful... The challenging thing is to find out where to extrapolate.**” Theorem 2 is providing a computationally feasible solution exactly to the question “**where to extrapolate**”. Thus we are confused by the reviewer’s new comments on ”Theorem 2 is not involved at all.”
> > >
> > > As the reviewer’s request on “what is the regret guarantee of BCOL?”, while we acknowledge theoretical guarantees are very insightful and can be used to guide online/offline RL algorithm design, we would argue that it is not the sole standard to judge RL algorithms and papers. We would refer reviewers to our experiment section, where our well-designed and comprehensive experiments clearly show the benefits of our  proposed method.
> > >
> > > There are many offline RL works with regret or sample complexity guarantees that are limited to linear function approximation settings [e.g. 3, and many other related work] without any empirical study. There are also few offline RL algorithms with theoretical guarantees, but without experiment on the challenging tasks in the D4RL benchmark [1,4,5]. Thus we think this paper still brings new contributions to the community.
> > >
> > > ### Computation cost
> > >
> > > We respectfully disagree with the reviewer that the computation cost is a major measure of offline RL algorithms. Almost all offline RL algorithms use one or more following tricks that increase the computation cost: introducing new regularizer terms, computing uncertainty sets, sampling actions multiple times from behavior policy, or taking conservative backup from multiple target Q networks. In addition to these common tricks, as more recent examples, the work cited in the review [1] and its prior work [5], introduced min-max objectives. Thus it introduces a whole new level of optimization problem and requires more computation. Another recent work [2] in offline RL introduced learning Q values, $Q(s,a,\delta)$, conditioned on a confidence level $\delta \in [0, 1]$. Offline RL algorithms are motivated by the scenarios where samples are costly and thus limited. Thus these algorithms are often motivated to to use samples more efficiently and thoroughly, with an additional cost of computation.
> > >
> > > We implement the Q function and policy with $B$ output heads in the last layer and vectorize the loop over $b$. Thus, the additional $O(B)$ computation cost only happens in the computation of loss and forward and backward pass of the last layer, rather than the whole network. Thus the additional $B$ factor does not apply to the whole amount of computation. (This is an exact implementation of Eq (6) and (7) without considering the sampling $b$ approach.)
> > >
> > > While our computation cost is comparable to other offline RL methods, we believe that it is not a major factor to consider when evaluating a new offline RL algorithm. In fact, we have not seen any work in offline RL that suggests that computation cost is a critical factor. Therefore, we do not believe that computation cost is relevant to the evaluation of our algorithm.
> > >
> > > Regarding the sample complexity introduced by $b$: There is no uncertainty or unknown transitions related to the variable $b$. It is unclear what the reviewer means by the possible sample complexity (in order to estimate any new random variable introduced by $b$).
> > >
> > > ### Response to the overall judgement
> > >
> > > **While the reviewer acknowledged the contribution of our algorithm, the main concerns are about why we did not make "other" contributions in the paper.** We found this unfair, as the reviewer did not identify any technical flaws, weak evaluation, inadequate reproducibility, or ethical considerations in our paper. These are the types of issues that would warrant a score of 3 per NeurIPS's definition of score 3. However, the reviewer assigned us a very low score without identifying any of these issues.
> > >
> > > Finally, it is important to note that while most offline RL methods rely on policy or value regularization, our work takes a fresh and new approach to offline RL which is very different from previous works and to the best of our knowledge, we are the first who propose such a fresh perspective in offline RL. This in itself demonstrates the significance of our work.

---

> > > > ### Author Response · Authors · 2023-08-13
> > > > **Reference for the last comment**
> > > >
> > > > [1] Bhardwaj, Mohak, et al. "Adversarial model for offline reinforcement learning." arXiv preprint arXiv:2302.11048 (2023).
> > > >
> > > > [2] Hong, Joey, Aviral Kumar, and Sergey Levine. "Confidence-conditioned value functions for offline reinforcement learning." arXiv preprint arXiv:2212.04607 (2022).
> > > >
> > > > [3] Jin, Ying, Zhuoran Yang, and Zhaoran Wang. "Is pessimism provably efficient for offline rl?." International Conference on Machine Learning. PMLR, 2021.
> > > >
> > > > [4] Yao Liu, Adith Swaminathan, Alekh Agarwal, Emma Brunskill "Provably good batch off-policy reinforcement learning without great exploration." Advances in neural information processing systems 33 (2020): 1264-1274.
> > > >
> > > > [5] Ching-An Cheng, Tengyang Xie, Nan Jiang, Alekh Agarwal. "Adversarially trained actor critic for offline reinforcement learning." International Conference on Machine Learning. PMLR, 2022.

---

> > > > ### Comment · Reviewer_bnLz · 2023-08-13
> > > >
> > > > Thank you again for your detailed response. I believe my biggest concern still remains.
> > > >
> > > > The authors first express their motivation in words  and subsequently framed an optimization problem which limits the time that a policy can deviate from the behavior policy. While there are many ways to limit the number of counterfactual explorations, the authors propose one specific way to do it in this work and proved that it is feasible to learn the optimal policy that follows the proposed constraint of counterfactual explorations. However, despite the authors' effort, I still fail to see the benefits brought by limiting the counterfactual explorations. **The authors at least need to formally show the gain comparing to not using counterfactual budgeting.** Without such insights, the benefits of the proposed method cannot be thoroughly appreciated, only staying on some common sense shared by the offline RL community. **Moreover, what is the benefit of the proposed method comparing to other offline RL regularization methods? Under what scenarios does the proposed method in this work outperform the other offline RL methods? Does it always outperform all other methods?**
> > > >
> > > > The authors have underscored for multiple times that the reviewer acknowledged the contribution of their work. But before these concerns are clear, the reviewer cannot acknowledge that the contribution is enough for recommendation.

---

> > > > > ### Author Response · Authors · 2023-08-13
> > > > > **Response to Reviewer bnLz**
> > > > >
> > > > > Thank you for engaging in a discussion with us. We hope that the remaining reviewer concerns can be addressed after these scholarly exchanges.
> > > > >
> > > > > > While there are many ways to limit the number of counterfactual explorations, the authors propose one specific way to do it in this work and proved that it is feasible to learn the optimal policy that follows the proposed constraint of counterfactual explorations.
> > > > >
> > > > > We are very confused with the reviewer's argument and objection here. Yes, we propose a very specific ( and novel) way to limit the number of counterfactual explorations with a proven guarantee and that is our contribution. Shouldn't it be like this? That said, we would love to hear about other existing ways in offline RL that do the very same thing (“limit the number of counterfactual explorations”) as we are not aware of such methods. The reviewer seems to suggest that there are many such methods, so we would appreciate it if you could provide us with some references.
> > > > >
> > > > >
> > > > > > However, despite the authors' effort, I still fail to see the benefits brought by limiting the counterfactual explorations. The authors at least need to formally show the gain comparing to not using counterfactual budgeting. Without such insights, the benefits of the proposed method cannot be thoroughly appreciated, only staying on some common sense shared by the offline RL community.
> > > > >
> > > > > We compared our method to all other recent offline RL methods where they do not use counterfactuals, and we show that our method is beneficial. That is exactly what the reviewer is asking. A well-designed and comprehensive experiment is also a formal and rigorous way to evaluate the algorithm. We wonder if the reviewer could please comment on the weakness of the evaluation and comparison between offline RL algorithms in paper. We'd be happy to address any specific concern.
> > > > >
> > > > >
> > > > > > Moreover, what is the benefit of the proposed method comparing to other offline RL regularization methods? Under what scenarios does the proposed method in this work outperform the other offline RL methods? Does it always outperform all other methods?
> > > > >
> > > > > Offline RL methods that rely on regularization are classified in two categories. Policy constraint-based methods only work when the data is collected using optimal or near-optimal policy methods, such as BCQ, BEAR, and DecisionTransformer. On the other hand, methods like CDC and CQL, which focus on value and policy constraints to address extrapolation, are more general and can work in most cases even if the behavior policy is not optimal. Our proposed method has the same property as them that it can work with data that is collected with any behavior policies. However, we formulate and address the problem of extrapolation in offline RL in a very different and novel way which hasn't been done before. Our experiments show that our method outperforms these methods on most of the environment, especially in challenging environments like AntMaze, where the data is collected from multiple policies and the agent must learn a policy that is a combination of these policies.
> > > > >
> > > > > We refer to section 3.2 and section 4 in our paper for more detailed discussion about comparing to other offline RL regularization methods.

---

> > > > > > ### Comment · Reviewer_bnLz · 2023-08-18
> > > > > >
> > > > > > My concern is finally resolved after seeing the example that the authors presented to Reviewer FoC4 to vividly demonstrate why budgeting works. Please include this example in the updated manuscript as it helps a lot with the understanding of the proposed method. I have raised my score accordingly.

---

### Official Review · Reviewer_FoC4 · 2023-07-03

**Soundness:** 3 good
**Presentation:** 3 good
**Contribution:** 3 good
**Rating:** 7
**Confidence:** 4

**Summary:**

This paper proposes a TD approach to induce counterfactual decisioning in offline RL agents. Basically, the approach suggests using a count-based budget that gives scope for making decisions that are not exhibited by the behaviour policy. The paper implements this approach in various standard benchmarks and shows that the method performs favourably to the state-of-the-art approaches. They also give a theoretical justification for why budgeting should work.

**Strengths:**

The strength of the paper lies in the simplicity of the proposed budgeting for counterfactual decision making. The paper identifies the gap in the existing approaches which fall short of inducing counterfactuals. The authors propose to put a hard budget on the number of counterfactual decisions taken by the agent. Building upon the constrained optimization formulation for solving the MDP, the work shows how it converges to a fixed point and argues about the optimality of the same. In addition, it gives a function approximation version for using deep learning based approaches in conjugation.

**Weaknesses:**

I find the following weaknesses in the approach:

**Tuning of the B**: It is unclear how the budget would be tuned for different environments. Because having a high B implies applying off-the-shelf RL algorithms without considering the nuances of offline RL setting, and a low B will make the method similar to imitation learning techniques.

**Lack of Imitation Learning baseline**: It would be helpful to see how the proposed BCOL compares with state-of-the-art approaches for imitation learning. This will highlight the importance of counterfactual decision-making learnt from offline dataset. I suggest authors include a competitive baseline for IL, too.

**Comment on the novelty of budgeting**: Safe RL approaches have count-based constraints on agent's violations of safety. Therefore, the theory given in the present work has a very high overlap with the prior works, reducing the novelty in the optimization or the fixed point derived.

**Motivation behind the counterfactual budgeting**: I am myself not aware of regularized techniques for increasing counterfactuals in offline RL, however from first principles, it seems that inducing counterfactuals might not be a good idea as offline RL might lead to safety issues when the agent is deployed. The aim of offline RL (at least to me) is finding connections in the offline data that can help improve the returns gained by taking care that the agent doesn't infer unrealistic novel behaviours from offline data that could turn harmful once deployed in the actual environment. Can authors please comment on the safety of the BCOL agents?

**Why would such budgeting work**: From equations 4 and Select() given on line 185, it looks like BCOL would spend the budget in the initial steps of decision-making, and after that, it would follow the behaviour policy. Put in a different way, BCOL's budgeting seems myopic in nature; it is unclear to me how the algorithm will induce the agent to use the budget later in the decision-making when it matters the most. Can authors please comment on how the budget would be used pragmatically by the agent? It would help to see an empirical analysis of the budget expenditure against the time steps.


**Questions:**

I have listed my questions and corresponding suggestions in the above section.

Overall, I like the paper's straightforward approach to increasing counterfactuals in offline RL and enjoyed reading the derivations provided. However, the paper has many unaddressed weaknesses, as pointed out above. I am inclined towards borderline rejecting the work in its current form. With my questions answered, I would love to increase my rating for this work.

**Limitations:**

Authors do point out the limitations around using count-based approach for budgeting and theoretical analysis of less number of counterfactuals. In addition, I urge authors to also talk about the safety considerations involved in BCOL.

---

> ### Author Rebuttal · Authors · 2023-08-10
>
> Thank you for your valuable feedback. We hope you will consider increasing your score after reading our responses. Please let us know if there are more questions.
>
> > “Tuning of the B”
>
> We agree with the reviewer on the algorithm’s behavior with high and low B. Such a behavior (spanning over the spectrum from imitation learning to vanilla RL, with different values of hyper-parameters) is a common and desired behavior for offline RL algorithms. We would like to restate that we use a single value of B across all Mujoco tasks, and another value of B across all AntMaze tasks. This shows the algorithm is less sensitive to hyperparameters than many offline RL algorithms that tune the hyperparameters per task. It also shows that it’s feasible to tune B over one simpler validation task that shares similar properties with the target task. We also want to remark compared with the coefficient of the pessimism/uncertainty regularization in other algorithms, B has a clear physical meaning and it is more intuitive, and is easier to be set by prior knowledge about the environment.
>
> > “Lack of Imitation Learning baseline”
>
> As the reviewer requested, we will include more imitation learning baselines in the paper. We include the results from BC on top 10% data, advantage weighted regression, and the decision transformer here, as the common imitation learning baselines in offline RL work. Please see the results in the rebuttal to all reviewers.
>
> > “the novelty of budgeting”
>
> We thank the reviewer for pointing out the connection between our work and count-based constraints violation in safe online RL. However, there is a key difference between our setting and safe online RL: in online RL, the agent can interact with the environment to collect more data, while in offline RL, only limited data is available and exploration is not an option. This difference has a significant impact on the way that we design methods for offline RL. Additionally, while count-based constraints violation has been studied in safe RL, to the best of our knowledge, our approach of using budget for counterfactual decisions in the context of offline RL is novel (also reviewer wQce pointed that out). Specifically, our proposed algorithmic idea of using dynamic programming to solve a constrained optimization problem in an offline setting has not been proposed before. We will include a discussion of the connections with safe RL in our paper. If the reviewer can provide a more specific reference that raises concerns about the novelty of our algorithm, we would be happy to include a more detailed discussion in the paper.
>
>
> > “Motivation behind the counterfactual budgeting”
>
> We would like to clarify that our algorithm is not “regularized techniques for increasing counterfactuals in offline RL”. In contrast, our algorithm can be viewed as *decreasing counterfactuals in offline RL*, as most offline RL methods do, but we are doing it in a simpler, more direct and explainable way. In short, RL methods are intrinsically learning over counterfactuals, and our budgeting idea is to upper bound it.
>
> More specifically, vanilla Q learning (or any off-the-shelf online RL method based on that), without any constraints, will update the Q values from the max Q values in the next time step:
> $$ Q(s,a) \leftarrow r(s,a) + \gamma \max_{a’} Q(s’,a’) $$
> This backup will implicitly update the greedy policy with a counterfactual action $\arg\max_{a’} Q(s’,a’)$ unless the argmax is behavior action. Online RL algorithms such as SAC, DDPG, TD3 are all based on learning these counterfactual actions. In the offline settings, a popular approach is adding a regularization term on top of the Bellman error on Q function, such as:
> $$ - (\max_{a’} Q(s,a’) - Q(s,a)) $$
> Where a is from the behavior policy. Such a regularization term prevents the Q values for counterfactual actions from being too large, thus it decreases the number of counterfactuals in the resulting Q function and its greedy policies. However, unlike our method, this type of popular regularized method cannot yield a test-time absolute upper bound of counterfactuals. In the sense of counterfactual decisions, our method is safer than other offline RL methods, except behavior cloning which follows behavior policy (factual decisions) anytime.
>
> Please see Section 3 in our paper for more discussion.
>
> > “Why would such budgeting work”
>
> We emphasize that our algorithm does not spend the budget in the initial steps of decision-making. The reason is that we use dynamic programming to solve the trade-off between spending the budget now to get a better action and keeping the budget for the future to get a better future value. Mathematically, it is described by the trade off between $\max_{a’} Q(s’, b-1, a’)$ and $E_{a’ \sim \mu} Q(s’,b,a’)$ in Eq (4) in our paper. The maximization in the first term represents the immediate benefit of greedy counterfactual actions, but the higher budget and thus higher Q values explains the benefit of taking a factual action for now to achieve a higher value from future counterfactual actions. Always taking the greedy at the beginning is myopic and not the optimal solution to this dynamic programming problem.
>
> In fact, our experiment shows there are still budgets left at the end of the episode during the test. We attached the total BCOL spent in the test, averaged over seeds and last 10 steps, for AntMaze tasks.
> Task | Average budget spent in test (max budget is 50) | Standard deviation of budget spent
> ---|---|---
> Antmaze-umaze-v0 | 41.00 | 0.91
> Antmaze-umaze-diverse-v0  | 20.02 |  6.44
> Antmaze-medium-play-v0 | 45.94 | 1.13
> Antmaze-medium-diverse-v0 | 46.04 |  0.92
> Antmaze-large-play-v0 | 43.33 | 1.81
> Antmaze-large-diverse-v0 | 43.75 | 1.77
>
> This result shows budget spent is less than 50, and there are budget left at the end of episodes. Thus the algorithm will not be forced to take behavior actions. In contrast, it will plan on how to spend the budget by line 1 in Select().

---

> > ### Comment · Reviewer_FoC4 · 2023-08-14
> >
> > Thanks for the detailed response to my review.
> >
> > Tuning of B:
> > The authors acknowledge that high and low B values will have the effects pointed out in the main review. However, I fail to understand why such behaviors are _desired_ in the context of the present work. The present work aims to induce counterfactuals in offline RL with count-based budgets. In that case, it would be great if authors shed more light on how B is chosen for a given environment; otherwise, issues with the value of B arise.
> >
> > Imitation Learning baseline:
> > Thanks for including the BC baseline. It answers my query.
> >
> > Novelty of budgeting:
> > The reason for pointing out the connection here is neither the present work nor the safe online RL works restrict themselves to offline or online settings, respectively, while deriving the fixed point theoretically. So, the authors can at least acknowledge the count-based optimization in safe RL. However, I acknowledge the novelty of applying count-based optimization for counterfactual induction in offline RL.
> >
> > Motivation behind the counterfactual budgeting:
> > Yes, the present work is not a "regularized technique." When I refer to them, I mean the previous works that try to induce counterfactual decision-making in offline RL. In this context, I request the authors to comment on the safety of inducing such counterfactuals, which might lead to unrealistic extrapolation and cause safety hazards when the offline-trained RL agent is deployed in the real world.
> >
> > Why would such budgeting work?:
> > I feel the table attached here answers the question of whether BCOL avoids myopic use of budget to a limited extent. I do agree with the authors that BCOL will limit the counterfactuals within the provided budget. However, my question is regarding the optimal use of the budget, i.e., empirically confirming whether the budget is used in the best way possible. I do get the argument that it is DP's task to ensure such strategic use of budget. But, it would still be helpful to provide at least a proof of concept on a small grid world. For simplicity, authors can verbally describe a 5x5 or some such grid world with few trajectories and concretely present a case for BCOL's budgeting.
> >
> > I am keeping the score for now, but I will be happy to continue the discussion on the above points.

---

> > > ### Author Response · Authors · 2023-08-15
> > > **Response to Reviewer FoC4's further questions (part 1)**
> > >
> > > We are pleased that you are onboard with the idea and merit of the paper. We hope our responses clarify the remaining issue and the reviewer will increase their scores as they already acknowledged the significance of our work.
> > >
> > > ### A meta clarification note about inducing counterfactuals
> > > > The present work aims to induce counterfactuals in offline RL
> > > > safety of inducing such counterfactuals
> > >
> > > We see this was repeated many times in the review and this response. Thus we want to clarify this to make sure we understand the reviewer’s comments correctly.
> > >
> > >
> > > All offline RL methods will introduce counterfactual decision-making as a nature of RL. It will be imitation learning if it does not plan on any counterfactual decisions. However, BCOL upper bounds this counterfactual in offline learning. We are not aware of any other offline RL works which directly upper bound the induced counterfactual decision-making in offline RL like us. Thus the goal of our work is not really **inducing counterfactuals** but **reducing counterfactuals**. It is the key insight and contribution of the whole paper and we really hope the paper expressed it clearly.
> > >
> > > ### Response to specific questions
> > > >Tuning of B: The authors acknowledge that high and low B values will have the effects pointed out in the main review. However, I fail to understand why such behaviors are desired in the context of the present work.
> > >
> > > That is a good question. The reason is offline RL requirements vary in different environments and dataset conditions. For example, the agent should act more conservatively while learning from a near-expert dataset, and it will require more exploration with a dataset collected by random policy. Thus the offline RL algorithm needs to be flexible to cover the spectrum from imitation learning (no exploration) to vanilla online RL (proactive exploration). Many existing offline RL algorithms share a similar property, e.g., IQL paper says “We also emphasize that our value learning method defines the entire spectrum of methods between SARSA ($\tau = 0.5$) and Q-Learning ($\tau \to 1$).”
> > >
> > > >The present work aims to induce counterfactuals in offline RL with count-based budgets. In that case, it would be great if authors shed more light on how B is chosen for a given environment; otherwise, issues with the value of B arise.
> > >
> > > We thank the reviewer for this good suggestion. For how we tuned B in our experiment, we discussed it on page 8 of that paper. For the suggestion about selecting B to the readers, we suggest starting from B=0 or 1 as an imitation-style baseline, then increasing or doubling the value of B which will add more RL components into the imitation learning baseline. We suggest stopping increasing the value of B after observing variance and instability during offline training. Thus we select the budgets to improve on top of imitation baseline and avoid over-extrapolate in offline RL. We will include more discussion in an updated version of our paper.
> > >
> > > > Imitation Learning baseline:Thanks for including the BC baseline. It answers my query.
> > >
> > > We thank the reviewer’s acknowledgement of our effort.
> > >
> > > > Novelty of budgeting
> > >
> > > We thank the reviewer for the good suggestion. We will include discussion about the count-based optimization in an updated version of our paper. Would it be possible if the reviewer points us to some specific references in safe RL to make sure we don’t miss the important ones?
> > >
> > > > In this context, I request the authors to comment on the safety of inducing such counterfactuals, which might lead to unrealistic extrapolation and cause safety hazards when the offline-trained RL agent is deployed in the real world.
> > >
> > > This is a good question. First, we must clarify that BCOL is not “inducing such counterfactuals”, and we hope it is not the underlying assumption behind the reviewer’s request. (See our meta clarification note). We want to make sure there is no misunderstanding on this point.
> > >
> > > We agree that the safety concern when the offline-trained RL agent is deployed in the real world is very important. We will include more discussion about that in the introduction and discussion section from that perspective. We also want to mention that the safe RL has very different setups than offline RL, though it overlaps with offline RL. Most existing offline RL have not been tested with the safety criteria formally, and it is not clear how they would perform. Our method is no exception.
> > >
> > > However, we identified that our algorithm design shares the similar motivation with the reviewer’s comments. Previous work, no matter how they regularize the value of policy function, cannot provide a guarantee about the distance between deployed policy and a safe baseline (behavior policy). For our algorithm, we have an absolute upper bound of violation of behavior policy during the test, if we can assume the behavior policy is always safe to execute. We would like to address further specific concerns about our algorithm on this aspect.

---

> > > ### Author Response · Authors · 2023-08-15
> > > **Response to Reviewer FoC4's further questions (part 2)**
> > >
> > > > Why would such budgeting work?
> > >
> > > We are glad that our responses addressed your questions.
> > >
> > > Here we provide a 3 x 4 grid-world example to illustrate how the budget and BCOL algorithm works. We describe the map below. S means starting state, F means a failed state with reward -1 and G means the goal state with reward 1. We refer to the position of state by (y, x) where (1, 1) stands for the top left corner (e.g. starting state S is (2, 1)). Star means empty grid (Openreview use markdown and it does not allow me to leave it empty.)
> > >
> > > —  —  —  — —  —
> > >
> > > | *  |  *  |  *  |  *  |
> > >
> > > —  —  —  —  —  —
> > >
> > > |S |  *  |  *  | G |
> > >
> > > —  —  —  —  —  —
> > >
> > > |  *  | F |  *  |  *  |
> > >
> > > —  —  —  —  —  —
> > >
> > > There are two types of trajectories in the dataset: type 1 and type 2 where most of the trajectories in the dataset are type 1, and there are only few type 2 trajectories. The trajectories are described below, with L, R, U, D denoting the four actions left, right, up, and down. E means the trajectories end.
> > >
> > > Trajectory type 1:
> > >
> > > —  —  —  —   —  —
> > >
> > > |R |D | * | * |
> > >
> > > —  —  —  —   —  —
> > >
> > > |U | R | R | E |
> > >
> > > —  —  —  —  —  —
> > >
> > > | * | * | * | * |
> > >
> > > —  —  —  —  —  —
> > >
> > >
> > > Trajectory type 2:
> > >
> > > —  —  —  — —  —
> > >
> > > | * | * | * | * |
> > >
> > > —  —  —  —   —  —
> > >
> > > |R | D | * | * |
> > >
> > > —  —  —  —  —  —
> > >
> > > | * | E | * | * |
> > >
> > > —  —  —  —  —  —
> > >
> > >
> > > Without any budget of counterfactuals, imitation learning agents will follow the first type of trajectories and take a longer path to the goal. With budget = 1, BCOL selects between following the empirical behavior policy (“U”) or an alternative action (“R”) in the starting state. Notice that there are a few type 2 trajectories, so that the high value of (2, 2) will be reflected in the backup value of action “R” in (2, 1). Thus taking action “R” and spending the budget results in a higher value. This is also because in the later states following trajectory type 1, all alternative actions will not lead to any state with a higher value than the goal state. The budget also prevents the agent from taking too many counterfactual actions e.g. in states (2,2) or (2,3), unless these counterfactual actions provide a higher reward gain than optimal path.
> > >
> > > With this type of grid-world examples, it is also obvious to see that we can change the state positions and trajectories so that the optimal way of spending the budget is in the middle of trajectories.
> > >
> > >
> > >
> > >
> > > > I am keeping the score for now, but I will be happy to continue the discussion on the above points.
> > >
> > > We are actually puzzled by the reviewer’s overall score as the major questions from the reviewer are about clarification and connection with safe RL work. We hope with the response above we can address the remaining questions. We are happy to address any concerns that prevent the reviewer from being able to increase the score.

---

> > > > ### Comment · Reviewer_FoC4 · 2023-08-15
> > > >
> > > > Thanks for the detailed clarifications. They satisfactorily answer my concerns. I am increasing my score to "Accept: 7".

---

### Official Review · Reviewer_wQce · 2023-07-05

**Soundness:** 3 good
**Presentation:** 4 excellent
**Contribution:** 3 good
**Rating:** 7
**Confidence:** 4

**Summary:**

This paper gives a total new solution to offline RL, instead of introduing pessimism by behavior constraint or value regularization, budgeting the number of counterfactual decisions, which naturally reduces overestimation. This paper also gives a good formulation of the problem and provides a nice solution to solve the allocation problem by dynamic programming. This paper shows good empirical performance over offline RL benchmarks and optimality of the fixed point solution as theoretical justification.

**Strengths:**

- The paper is very well written.
- The idea of avoid distributional shift by budgetting the number of counterfactual decision making is very novel and promising.
- This paper gives a nice formulation of the problem and provides a solution to solve the allocation problem by dynamic programming, which is also solid and theoretical justificated.
- This paper shows strong performance on various D4RL datasets, though not SOTA in some datasets if compared to more recent works.

**Weaknesses:**

- This method introduces two hyperparameters to tune: $\omega$ and $B$.
- It is better to have some visualization on which critical steps does the algorithm tends to make counterfactual decisions and whether it has some physical meanings. For example, on antmaze tasks, which location does the algorithm make counterfactual decisions.

**Questions:**

- Why adding the gap_penalization term (Equation 9) is necessary? Does it still do some kind of behavior regularization implicitly? Could you elabotate more on the usage of Equation 9?
- Why SAC+BCOL can work well on antmaze tasks while TD3+BCOL can't?

**Limitations:**

See Weakness.

---

> ### Author Rebuttal · Authors · 2023-08-10
>
> We thank the reviewer for their detailed and positive feedback. Please see our responses to your comments and let us know if there are more questions.
>
> > This method introduces two hyperparameters to tune
>
> That is a fair point. Our method does have two hyperparameters, but this does not diminish its efficacy or applicability as a practical offline RL algorithm. In fact, even without the hyperparameter $\omega$ ($\omega=0$), our method (BCOL with SAC implementation) shows strong performance (and sometime remains state-of-the-art) in Mujoco and AntMaze tasks, as shown in Figure 1 and Table 6 of our paper. However, the issue of hyperparameters is an open problem in offline RL, and other methods such as IQL, CQL, and CDC also have various hyperparameters.
>
> > More visualization on critical steps and counterfactual decisions
>
> That is a good suggestion, thank you. We will work on creating visualizations for the camera-ready version of our paper to show which critical steps the algorithm tends to make counterfactual decisions in AntMaze tasks.
>
> >Why adding the gap_penalization term (Equation 9) is necessary:
>
> That is a good question. $Q(s,b+1,a) \ge Q(s,b,a) $ is a property that must be satisfied for our desired function $Q$. However, such a property is not always held when we optimize the TD errors. Thus enforcing such constraints can be helpful to learn the desired Q function. One way to enforce it is to design neural network architectures outputting a monotonically increasing sequence of real numbers, as $Q(s, \cdot, a)$, but it comes with additional complexity of the network structure and training. Thus we choose a simpler way of adding a regularization term to control the violence to property $Q(s,b+1,a) \ge Q(s,b,a) $. Such a regularizer does not directly enforce a behavior constraint to the policy. Because this regularizer constraints Q gaps for the same action with different budgets, instead of gaps between different actions. But this regularizer will help the dynamics programming over counterfactual actions finding a self-consistent solution. We will include more discussion about this in the corresponding section of the updated version.
>
> >Why SAC+BCOL can work well on AntMaze tasks while TD3+BCOL can't?
>
> That is a great question. We believe this is an issue with TD3 algorithm and offline RL based on TD3 in general. For example, the original TD3+BC paper does not report on AntMaze tasks. Our reproduced results show TD3+BC perform poorly on AntMaze. Similarly, prior work [1] observed BCQ (another TD3 based offline RL algorithm) perform poorly either on AntMaze. In the meantime, we noticed there are fewer TD3 based offline RL algorithms, in comparison with SAC based architecture. Thus our hypothesis is that TD3 based algorithms face larger challenges in offline RL especially in tasks like AntMaze where the offline algorithm needs to find a policy that stitches together multiple policies to find the optimal policy.
>
> [1] Fakoor, Rasool, et al. "Continuous doubly constrained batch reinforcement learning." Advances in Neural Information Processing Systems 34 (2021): 11260-11273.

---

### Official Review · Reviewer_4X4f · 2023-07-09

**Soundness:** 3 good
**Presentation:** 3 good
**Contribution:** 2 fair
**Rating:** 4
**Confidence:** 2

**Summary:**

This paper designs an new algorithm for offline RL. It also conduct experiment to validate its algorithms.

**Strengths:**

The algorithm proposed is new. The intuition behind is presented clearly.

**Weaknesses:**

1. The performance of the proposed algorithm does not demonstrate substantial superiority compared to the Contrastive Q-Learning (CQL) algorithm. It would be beneficial to provide more persuasive evidence of the proposed algorithm's efficacy.

2. The use of the "budget" concept presents limitations, as it appears applicable primarily in tabular settings. Its adaptability to continuous cases is challenging, which curtails its generalizability. The authors may want to consider elucidating the feasibility of this concept in a broader context or proposing alternative approaches for continuous settings.

**Questions:**

The performance of the proposed algorithm does not demonstrate substantial superiority compared to the Contrastive Q-Learning (CQL) algorithm. It would be beneficial to provide more persuasive evidence of the proposed algorithm's efficacy.


**Limitations:**

The use of the "budget" concept presents limitations, as it appears applicable primarily in tabular settings. Its adaptability to continuous cases is challenging, which curtails its generalizability. The authors may want to consider elucidating the feasibility of this concept in a broader context or proposing alternative approaches for continuous settings.

---

> ### Author Rebuttal · Authors · 2023-08-10
>
> Thank you for your valuable feedback. We hope you will consider increasing your score after reading our responses. Please let us know if there are more questions.
>
> > The performance of the proposed algorithm does not demonstrate substantial superiority compared to the Contrastive Q-Learning (CQL) algorithm.
>
> Contrary to what the reviewer suggested, our results are better than the reported results with CQL. In particular, for the total score in Mujoco control tasks (746 vs 713) and, more importantly, the harder AntMaze tasks (396 vs 352.9), our algorithm BCOL surpasses the CQL performance by a large margin. (Please see the discussion about why AntMaze are more meaningful benchmark in our response to Reviewer bnLz about “comparison with CQL and CDC”.) We consider it a substantial improvement given the merits of CQL and how well it is performed in Mujoco tasks (e.g. a contemporaneous work [1] summarized the comparison between CQL and several more recent offline RL algorithms).
>
> We would like to bring the reviewer's attention to two observations about how CQL results are mainly reported in related works. In short, we tried our best to be as fair as possible to CQL, which is different from the way CQL is reported in literature. Thus the improvement on top of CQL does not look as large as in prior work.
> 1. CQL website [2] keeps updating its performance and the performance has been significantly improved after the publication of the paper, especially after the D4RL Mujoco-v2 environments. In the meantime, many prior works [3,4,5,6] only compare to the results (in Mujoco-v0) in the original CQL paper. That explains why the gap with CQL in these papers is so large.
> 2. In our submission, we reported CQL’s results on AntMaze environments from the CQL paper. However, many recent papers [1,7,8,9] compare proposed methods against their reproduced CQL results, not the original numbers in CQL. Their reproduced CQL results are often worse than the original numbers in CQL. E.g. offline results of CQL in Table 2, and Appendix C in [7] say “Our reproduced results offline are worse than the reported results, particularly on medium and large AntMaze environments.” Table 1 in [1], Table 1 in [8] and Footnote 4 on Page 8 in [9] report the results similarly. That explains why the gap with CQL in other papers is very large.
>
>
> > The use of the "budget" concept presents limitations, as it appears applicable primarily in tabular settings
>
> We believe there is a misunderstanding. We would like to clarify that we derive a practical Bellman operator designed for function approximation in Section 3.1. Our method works with continuous state and continuous action spaces, as evidenced by our comprehensive experiments with gym MuJoCo and AntMaze in the D4RL benchmark. Both of these environments have continuous state and action spaces. Therefore, we are unsure of what the term "tabular" refers to in our context.
>
> If the review means the budget space is discrete, we would acknowledge this in our current method. However, this is not a fundamental limit for the general “budget” method. We can define a continuous distribution distance metrics as a budget variable, and as an input to the Q function $Q(s,a,b)$. The sum over $b$ in training loss can be replaced with sampling $b$ uniformly. We would like to emphasize that our method itself shows the effectiveness of budgeting ideas, and we left the continuous budgeting implementation as a future work. We will include more discussion about this in the Discussion section in an updated version.
>
> [1] Improving and Benchmarking Offline Reinforcement Learning Algorithms. Kang et al. 2023
>
> [2] https://sites.google.com/view/cql-offline-rl
>
> [3] Chen, Lili, et al. "Decision transformer: Reinforcement learning via sequence modeling." Advances in neural information processing systems 34 (2021): 15084-15097.
>
> [4] Brandfonbrener, David, et al. "Offline rl without off-policy evaluation." Advances in neural information processing systems 34 (2021): 4933-4946.
>
> [5] Cheng, Ching-An, et al. "Adversarially trained actor critic for offline reinforcement learning." International Conference on Machine Learning. PMLR, 2022.
>
> [6] Bhardwaj, Mohak, et al. "Adversarial model for offline reinforcement learning." arXiv preprint arXiv:2302.11048 (2023).
>
> [7] Kostrikov, Ilya, Ashvin Nair, and Sergey Levine. "Offline reinforcement learning with implicit q-learning." arXiv preprint arXiv:2110.06169 (2021).
>
> [8] Fakoor, Rasool, et al. "Continuous doubly constrained batch reinforcement learning." Advances in Neural Information Processing Systems 34 (2021): 11260-11273.
>
> [9] Fujimoto, Scott, and Shixiang Shane Gu. "A minimalist approach to offline reinforcement learning." Advances in neural information processing systems 34 (2021): 20132-20145.

---

> ### Author Response · Authors · 2023-08-15
> **Thank you**
>
> We thank the reviewer for their time and effort. As the discussion period goes, we would be happy to explain anything further or address more questions. If we were able to address your questions and concerns, then we would appreciate if you can update your review. Thanks again for putting the time into reviewing our paper.

---

### Author Rebuttal · Authors · 2023-08-10

We thank the reviewers for their feedback that we've used to greatly improve the paper. We have responded to the concerns of the reviewers as individual comments below.
We are glad that the reviewers found that our method is simple and straightforward (FoC4), novel (4X4f, wQce, bnLz), and is a solid approach and theoretically justified method (wQce). They also said that the paper generally is well-written, an enjoyable reading, presents a clear message, and the intuition behind is presented clearly (4X4f, wQce, bnLz). They found our experiments to be clear and comprehensive (wQce, bnLz) with strong performance (wQce).
We would like to clarify some key facts and summarize our main contributions here:

1) To the best of our knowledge, this is the first work to upper bound the number of counterfactual decisions in the context of offline RL. Such an objective is more straightforward, and explainable than many other offline RL methods.

2) We propose a dynamic programming approach to optimize this objective. By planning on the benefit from different extrapolation steps, our algorithm will balance between taking immediate greedy action to Q values, and the potential benefit of taking a more advantaged action in the future at the opportunity cost.
3) Theoretically, we prove our dynamic programming algorithm finds asymptotically the optimal way to allocate the budget of counterfactual, given an upper bound of counterfactual decisions.
4) Through our comprehensive experimental results, we show that our algorithm outperforms most of the offline RL baselines. It verifies the effectiveness of bounding the counterfactual decisions in offline RL.

We have included new experimental results in response to the reviewer comments from recent papers, and these results show that our method outperforms the latest SOTA by a large margin on the AntMaze task (one of the harder tasks in the D4RL benchmark) and is comparable to latest SOTA in Mujoco tasks. These latest results further strengthen our claims about the effectiveness of applicability of our method.

We attach the comparison against imitation learning baselines here.

Task|10% BC| AWR | DT | BCOL
---|---|---|---|---
Mujoco total | 666.2 | 308.5 | 672.6 | 746.0
AntMaze total | 134.2 | 126.3 | 112.2 | 396.0

For more recent offline RL baselines, we compare BCOL against them here as well.

Tasks | ARMOR| MoRel | MOPO | RAMBO | COMBO  | ATAC | MuZero | CRR+ | CQL+| BCOL
--- | --- | --- | --- | --- | --- | --- | --- | --- | --- | ---
 Mujoco total | 788.6  |  656.5  |  379.3  |  753.1  |  738.3  |  792.4  |  140.2  |  703.9  |  717.9 | 746.0
AntMaze total | -  |  0  |  0   | 37.8  |  137.6 |   -  |  0  |  41.9  |  89 | 396.0

As these results show, BCOL outperforms IL baselines with large margins. BCOL is comparable to the latest SOTA in Mujoco tasks, and is substantially better in AntMaze tasks. For simplicity, we only list the total score here for all new results in rebuttal, but we will include the full results in the paper.

---

### Decision · Program_Chairs · 2023-09-21

**Decision:**

Accept (poster)

**Comment:**

All the reviewers agree that the paper contributes a novel, interesting and effective solution approach to the well-motivated problem of offline reinforcement learning. The authors included additional experiments (comparison to imitation learning baselines, another gridworld domain, and analysis of budget consumption) that substantially strengthen the paper and should be included in the revision.